# Kinesin-1 conformational dynamics are controlled by a cargo-sensitive TPR switch

Shivam Shukla[1,2†], Jessica A Cross[1,2,3†], Monika Kish[4,5†], Sathish KN Yadav[1], Johannes F Weijman[1], Laura O'Regan[1], Judith Mantell[1], Ufuk Borucu[6], Xiyue Leng[2], Christiane Schaffitzel[1], Jonathan J Phillips[4,5*], Derek N Woolfson[1,2,7*], Mark P Dodding[1*]

[1]School of Biochemistry and Biomedical Sciences, University of Bristol, Biomedical Sciences Building, University Walk, Bristol, United Kingdom; [2]School of Chemistry, University of Bristol, Cantock's Close, Bristol, United Kingdom; [3]School of Engineering Mathematics and Technology, University of Bristol, Bristol, United Kingdom; [4]Living Systems Institute, University of Exeter, Stocker Road, Exeter, United Kingdom; [5]Department of Biosciences, University of Exeter, Stocker Road, Exeter, United Kingdom; [6]GW4 Facility for High-Resolution Electron Cryo-Microscopy, University of Bristol, Bristol, United Kingdom; [7]Max Planck-Bristol Centre for Minimal Biology, University of Bristol, Cantock's Close, Bristol, United Kingdom

*For correspondence:
jj.phillips@exeter.ac.uk (JJP);
D.N.Woolfson@bristol.ac.uk
(DNW);
mark.dodding@bristol.ac.uk
(MPD)

†These authors contributed
equally to this work

Competing interest: The authors
declare that no competing
interests exist.

Reviewing Editor: Julien Roche,
Iowa State University, United
States

## eLife Assessment

The revised manuscript by Shukla et al. provides **important** mechanistic insights into kinesin-1 auto-inhibition and cargo-mediated activation. Through a **convincing** integration of protein engineering, computational modeling, biophysical assays, HDX-MS, and electron microscopy, the study delineates how cargo binding induces an allosteric transition that propagates along the coiled-coil stalk to the motor domains, enhancing MAP7 engagement. The revisions substantially improve clarity, figure annotation, and methodological transparency, leaving the remaining limitations, primarily those inherent to conformational heterogeneity and resolution, appropriately acknowledged. Overall, the updated manuscript presents a coherent mechanism for kinesin-1 activation that will be of broad interest to the motor protein, structural biology, and cell biology communities.

**Abstract** Kinesin-1 is a dynamic heterotetrameric assembly of two heavy and two light chains (KHC and KLC) that mediates microtubule-based intracellular transport of many different cargoes. The complex adopts a compact, autoinhibited state that is activated by cargo-adaptor proteins containing specific short linear peptide motifs (SLiMs). These motifs interact with the tetratricopeptide repeat (TPR) domains of the KLCs. The mechanism coupling SLiM recognition to activation-associated conformational changes in the complex is unknown. Here, we combine protein design, computational modelling, biophysical analysis, and electron microscopy to examine the structural and mechanistic consequences of SLiM binding to the KLC-TPR domain within the complete heterotetrameric holoenzyme. We show that coiled coil 1 (CC1) of the KHC docks KLC TPR domains in the autoinhibited complex, forming the 'shoulder' feature observed in electron microscopy. Disrupting this interaction or binding an activating SLiM dislocates the TPR shoulder, freeing the motor domains and promoting transition between its closed, inactive, and open states. Opening the kinesin-1 complex facilitates binding to the microtubule-associated kinesin-1 cofactor, microtubule-associated protein 7 (MAP7). Therefore, cargo-mediated dislocation of the TPR

shoulder serves as a key initial step in kinesin-1 activation, allosterically linking cargo binding to motor dynamics.

## Introduction

Intracellular transport relies on the conformational dynamics of cytoskeletal motor proteins. One such process involves ATPase-driven binding and movement along cytoskeletal tracks. Another, less-understood process involves shape changes in motor complexes, which regulate movement in response to inputs like cargo or regulatory protein binding, ensuring precise transport control (*Yildiz, 2025*; *Cross and Dodding, 2019*; *Cross, 2016*; *Reck-Peterson et al., 2018*). Conformational transitions between a compact inhibited state and an open active state are essential for the regulation of kinesin-1, a ubiquitous and prototypic family of microtubule motors involved in transporting proteins, ribonucleoproteins, vesicles, and organelles in cells (*Yildiz, 2025*; *Cross and Dodding, 2019*; *Vale et al., 1985*; *Hackney et al., 1992*; *Hisanaga et al., 1989*; *Hirokawa et al., 1989*). Kinesin-1 is also hijacked by pathogens during infection, and its dysregulation is linked to neurological disorders (*Dodding and Way, 2011*; *Sleigh et al., 2019*).

Heterotetrameric kinesin-1 consists of two kinesin heavy chains (KHCs) and two kinesin light chains (KLCs). In mammals, KHCs are encoded by three paralogs—KIF5A, KIF5B, and KIF5C—while KLCs are encoded by four paralogs (KLC1–KLC4) (*Xia et al., 1998*; *Rahman et al., 1998*; *Gauger and Goldstein, 1993*; *Fan and Amos, 1994*). The KHCs have ATPase motor domains at the amino terminus, followed by coiled-coil domains (CC0–CC4) that mediate KHC dimerisation. CC0 forms the neck between the amino-terminal motor domains and the coiled-coil stalk which comprises CC1 – CC4. The carboxy-terminus contains an unstructured domain with the autoinhibitory 'IAK' motif, which regulates ATPase activity in the compact state, but is not essential for its formation (*Hackney, 2007*; *Kaan et al., 2011*; *Dietrich et al., 2008*; *Hackney et al., 2009*; *Friedman and Vale, 1999*; *Coy et al., 1999*; *Atherton et al., 2024*; *Tan et al., 2023*; *Yang et al., 1989*). The KLCs, which also help to regulate kinesin-1 activity, have a coiled-coil domain that binds to CC3 of the KHCs, followed by an unstructured linker to a tetratricopeptide repeat (TPR) domain critical for cargo binding (*Cross and Dodding, 2019*; *Yip et al., 2016*; *Verhey et al., 1998*; *Verhey et al., 2001*; *Figure 1A*). In several KLC isoforms, the TPR domain is followed by an intrinsically disordered domain, which includes a membrane-binding amphipathic helix that can stabilise interactions with membranous cargoes (*Antón et al., 2021*; *Cyr et al., 1991*; *Woźniak and Allan, 2006*).

Kinesin-1 activity is regulated by its conformational state (*Hackney et al., 1992*; *Hisanaga et al., 1989*; *Hirokawa et al., 1989*; *Yip et al., 2016*; *Cai et al., 2007*; *Stock et al., 1999*). Recent studies using negative stain electron microscopy (NS-EM), AlphaFold2 modelling, small-angle X-ray scattering (SAXS), and cross-linking mass spectrometry have revealed the architecture of the compact, autoinhibited state, named the 'lambda particle' (*Tan et al., 2023*; *Weijman et al., 2022*; *Carrington et al., 2024*). Key features of this particle include the ATPase motor domains, forming the 'head,' and a flexion point in the coiled-coil scaffold between CC2 and CC3—the 'elbow'—that enables the autoinhibited conformation to form (*Tan et al., 2023*; *Weijman et al., 2022*; *Carrington et al., 2024*). This flexion is functionally significant, as the complex can be activated or stabilised in its inhibited state through targeted mutations or binding of de novo designed peptide-based activators (*Weijman et al., 2022*; *Cross et al., 2024*). NS-EM and SAXS also reveal a 'shoulder' near the motor domains. Domain-deletion experiments indicate that the shoulder is formed by at least one KLC TPR domain (*Weijman et al., 2022*). Cross-linking mass spectrometry, NS-EM, and AlphaFold2 modelling suggest that the motor domains invert relative to CC1, binding back upon it and interacting with CC4 in a hierarchical inhibition mechanism (*Tan et al., 2023*; *Carrington et al., 2024*) and propose many possible contacts between the KLC-TPR domains with the motor domains and coiled-coil body (*Tan et al., 2023*). Key features of the inhibited complex appear to be conserved across the closely related mammalian KHC paralogues (*Tan et al., 2023*; *Weijman et al., 2022*; *Carrington et al., 2024*).

Understanding the position and function of TPR domains in the inhibited state is crucial for uncovering how kinesin-1 is activated, as these domains provide binding sites for activating cargo adaptors with W-acidic SLiMs (*Cross and Dodding, 2019*; *Pernigo et al., 2013*; *Cross et al., 2021*). W-acidic motifs bind the concave groove of the TPR superhelix, increase its curvature, and displace an autoregulatory interaction with the CC-TPR linker region to modulate KLC conformational state

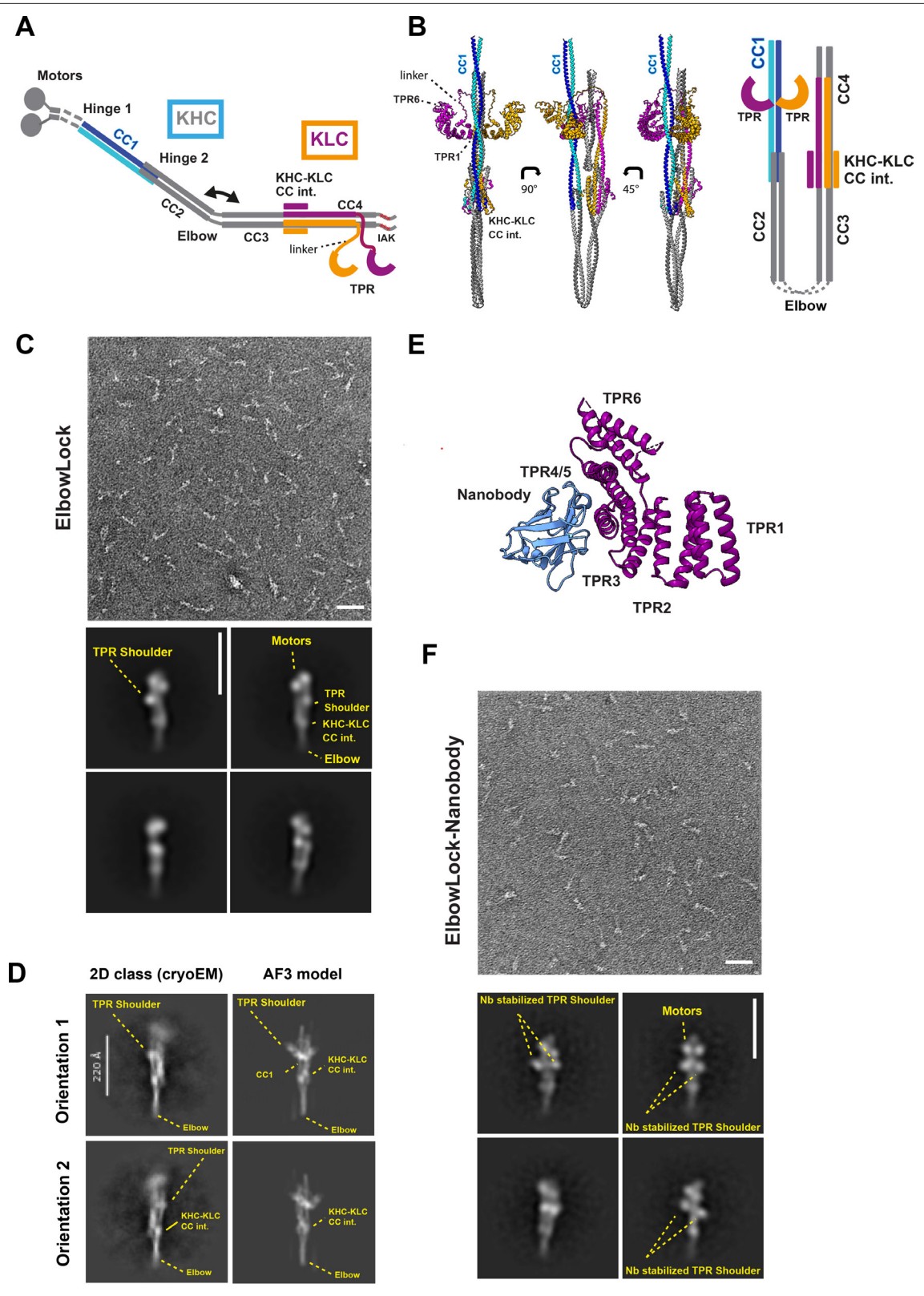

**Figure 1.** Computational modelling and single particle electron microscopy (EM) analysis identify a tetratricopeptide repeat (TPR) docking site (TDS) on CC1. (**A**) Schematic representation of heterotetramer kinesin-1 in an open conformation. (**B**) AF3 model in three orientations alongside a schematic showing kinesin heavy chain (KHC) (KIF5C) coiled-coil assembly. KHCs are shown in blue and cyan, KLCs in orange and purple. The complete AF3 output, including models for the full tetramer and sequences used are described in Figure S1. The schematic shows predicted domain positioning,

*Figure 1 continued on next page*

*Figure 1 continued*

omitting linker sequences and separating the KHC coiled-coils for clarity (**C**) Representative NS electron micrograph (from three independent experiments) showing ElbowLock complexes (scale bar is 50 nm) with selected 2D classes from negative stain electron microscopy (NS-EM) (scale bar is 26 nm). (**D**) Reference-free 2D class averages showing two orientations of complete heterotetrameric kinesin-1 complexes, compared to low-pass filtered back-projection of the AF3 model (scale bar is 22 nm). (**E**) Structure of the nanobody-TPR complex highlighting binding on exterior of TPRs 4 and 5 (pdb:6fuz). Cargo JIP1 Y-acidic peptide was removed for clarity. (**F**) Representative NS electron micrograph showing ElbowLock-nanobody complexes (scale bar is 50 nm) with selected 2D classes from NS-EM (scale bar is 26 nm).

The online version of this article includes the following source data and figure supplement(s) for figure 1:

**Figure supplement 1.** AlphaFold3 models of KHC-KLC complexes.

**Figure supplement 2.** Representative samples of protein complexes used in this study.

**Figure supplement 2—source data 1.** Labelled full gel images relating to *Figure 1—figure supplement 2*.

**Figure supplement 2—source data 2.** Raw files for Coomassie gel images relating to *Figure 1—figure supplement 2*.

**Figure supplement 3.** Negative stain electron microscopy (NS-EM) workflow.

**Figure supplement 3—source data 1.** Labelled full gel images relating to *Figure 1—figure supplement 2*.

**Figure supplement 3—source data 2.** Raw files for Coomassie gel image relating to *Figure 1—figure supplement 2*.

**Figure supplement 4.** Representative sets of 2D classes for all complexes.

(*Yip et al., 2016*). This molecular switch is also the target for the kinesin-1 activating small-molecule kinesore (*Randall et al., 2017*). Adaptors carrying W-acidic motifs include the lysosomal adaptor SKIP (*Pernigo et al., 2013*; *Rosa-Ferreira and Munro, 2011*; *Dodding et al., 2011*), the nuclear envelope adaptors Nesprin-2/4 (*Chiba et al., 2022*; *Wilson and Holzbaur, 2015*), and the amyloid-precursor protein (APP) vesicle adaptor calsyntenin-1 (*Kawano et al., 2012*; *Araki et al., 2007*; *Konecna et al., 2006*), and others (*Cross and Dodding, 2019*). Recently, we have designed a small-peptide ligand, KinTag, for the TPR binding site. This combines features of natural micromolar affinity W-acidic and related Y-acidic motifs to give high-affinity (low nanomolar) binding through simultaneous occupation of the W-, Y-, and F- pockets on the TPR receptor (*Verhey et al., 2001*; *Pernigo et al., 2018*; *Nguyen et al., 2018*; *Cross et al., 2021*). Notably, KinTag drives changes in TPR conformation like the natural ligands. Therefore, KinTag provides a useful tool to probe the mechanistic consequences of stable motor-cargo interactions, which in the natural system are also enabled by other co-operative protein-protein and protein-lipid interactions (*Antón et al., 2021*; *Sanger et al., 2017*). Importantly, when fused to an integral membrane protein, KinTag, or its parental W-acidic sequence, is sufficient to drive kinesin-organelle recruitment and axonal transport (*Cross et al., 2021*; *Kawano et al., 2012*; *Pu et al., 2015*; *Farías et al., 2015*). Thus, structural and biophysical analysis of isolated TPR domains in vitro and functional analysis of organelle transport in cells demonstrates a key role for SLiM-TPR binding in motor activation and has defined some important components of the pathway. However, the mechanism that couples SLiM cargo recognition to activation-linked conformational change in the complete kinesin-1 holoenzyme remains unknown.

To define this mechanism, we use protein design and engineering approaches to overcome intrinsic conformational dynamics and low-affinity motor-cargo interactions to stabilize intermediates along the kinesin-1 inhibition-activation pathway, enabling their biochemical characterisation and visualisation. We show that the KLC TPR domains dock onto CC1 in the autoinhibited complex to form the shoulder. Disrupting this interaction or binding an activating SLiM dislocates the shoulder. This makes the motor domains more accessible and dynamic as judged by hydrogen-deuterium exchange measurements in the open, active, and cargo-bound states compared with the closed, inactive form. Opening the complex also promotes interaction with the microtubule-associated cofactor, MAP7. Therefore, cargo-mediated dislocation of the TPR shoulder serves as a key step in kinesin-1 activation, allosterically linking cargo recognition to conformational dynamics.

## Results

## Coiled coil 1 (CC1) provides a TPR docking site (TDS) in the autoinhibited kinesin-1 complex

To begin, we sought to clarify the position of the TPR domains in the lambda particle by modelling their potential for associations using AlphaFold3 (AF3). AF3-generated assemblies of the hetero-tetrameric complexes of the KHC coiled coils (KIF5C) and KLC (KLC1A), that included the TPR domains (KIF5C/KLC1A) yielded models that were invariably folded-over at the elbow, with a global coiled-coil architecture consistent with our previous work (*Weijman et al., 2022*; *Cross et al., 2024*; *Figure 1B*, *Figure 1—figure supplement 1A*). Interestingly, the two TPR domains were confidently predicted to bind with C2 symmetry on CC1 'end on' via the first repeats (TPR1) at a position concurrent with the shoulder observed in NS-EM. This TPR conformation closely resembles their known mode of binding to the coiled-coil leucine zipper motif of the cargo adaptor JIP3 (*Cockburn et al., 2018*). The concave cargo binding surface on the TPR domain that recognises SLiM adaptors was typically occupied by the linker region that connects the TPR to the KHC-KLC coiled-coil inter-face (*Yip et al., 2016*). Models of the complete heterotetramer, including the motor domains were similar. The motor domains were inverted around Hinge 1, enabling interactions between them and the first part of CC1 and the end of CC4 with the neck coiled coil (CC0) projecting away from the motors, in line with recent reports (*Tan et al., 2023*; *Carrington et al., 2024*; *Figure 1—figure supplement 1B and C*).

To test these predictions experimentally, we examined purified kinesin-1 complexes using negative stain electron microscopy (NS-EM) followed by 2D classification of single particles (*Figure 1—figure supplements 2–4*). We focused on the ElbowLock background. This KIF5C construct contains a short, 5-amino acid deletion that restricts flexibility around the elbow and helps maintain particles in their lambda conformation, providing homogenous samples and facilitating subsequent analysis (*Cross et al., 2024*). Proteins were also stabilised using the amine-to-amine crosslinker BS3 that was important for achieving reproducibly high-quality samples for imaging. As observed previously for closed wild-type particles, the TPR 'shoulder' feature was readily apparent and typically appeared as a single globular density proximal to the two motor domains (*Tan et al., 2023*; *Weijman et al., 2022*; *Figure 1C*).

To visualise finer structural details, we turned to single-particle cryo-EM analysis of frozen-hydrated samples. We were unable to obtain optimal samples suitable for determining the complete structure. Nonetheless, we obtained reference-free 2D class averages that appeared to show full-length 'side' views of the complex with clear definition of the elbow, hinge 2, and KHC-KLC (coiled-coil) interface features (*Figure 1D*). The motor domains were poorly resolved in these classes, suggesting that the head assembly is somewhat flexible relative to the coiled-coil/TPR body. A comparison to low-pass filtered back-projections from the AF3 model (without motor domains) revealed density at a position concurrent with the docked TPR domains (*Figure 1D*).

However, TPR domain stoichiometry at the shoulder remained unclear. We reasoned that if the AF3 models were correct, then the latter TPR repeats (*Reck-Peterson et al., 2018*; *Vale et al., 1985*; *Hackney et al., 1992*) would be positioned away from the interface with the CC1. The KLC TPR domains are small and known to be conformationally dynamic, and so this might make it difficult to resolve them in 2D class averages (*Yip et al., 2016*; *Pernigo et al., 2013*; *Pernigo et al., 2018*). To test this hypothesis, we assembled complexes with a nanobody to bind and stabilise TPRs 4 and 5 and add additional mass (13 kDa per TPR) (*Cross et al., 2021*; *Pernigo et al., 2018*; *Figure 1E*, *Figure 1—figure supplement 2*). This resulted in NS-EM 2D classes with density extending either side of the coiled-coil scaffold that were fully consistent with the AF3 tetramer models and binding of both TPR domains on CC1 (*Figure 1E and F*). There was no evidence of additional density else-where in the complex. Thus, AF3 predicts a model for TPR domain association with the coiled-coil scaffold that is supported by electron microscopy experiments. Because we have used a stabilisation approach, we do not exclude the possibility that there may be differences in TPR stability, dynamics, and occupancy at the two symmetrical CC1 sites in the absence of stabilisation. i.e., only one TPR may be typically bound at one time, and/or they may have potential for exchange. Hereafter, the predicted TPR docking site on CC1 is called the TDS.

## Removal of the TDS dislocates the kinesin-1 shoulder

To test the role of the TDS in positioning the KLC TPRs directly, we removed it while preserving the overall length and coiled-coil structure of CC1. To do this, four heptad repeats of CC1 of KIF5C, flanking and incorporating TDS, were replaced with a de novo-designed homodimeric coiled coil, CC-Di (*Fletcher et al., 2012*). The aim was to retain the dimeric coiled-coil structure and length of CC1 while eliminating side chains that promote its interface with the TPR (ΔTDS) (*Figure 2A*). Supporting this, models of ΔTDS complexes using AF3 showed the expected seamless insertion of CCDi into CC1, with displacement of the TPR domains to a variety of different positions, in 5 models, all with high position error with respect to KHC (*Figure 2—figure supplement 1*). We used size-exclusion chromatography (SEC) to further validate our design. ElbowLock proteins elute as a single lambda peak, and this elution profile was unchanged by the ΔTDS modification, indicating that complex assembly or folding is not disrupted (*Figure 2B*). ElbowLock-ΔTDS complexes were then analysed by NS-EM and 2D classification (*Figure 2C and D*) and quantitatively compared to the parental ElbowLock (*Figure 2E*). The shoulder was lost in the ΔTDS background, suggesting that TPR domains had undocked from the core of the complex.

## The TDS binds directly to the KLC TPR domain

To determine whether the TPR binds TDS directly, we measured the interaction between fluorescently labelled synthetic peptides corresponding to the TDS region and the TPR domain directly using fluorescence polarisation (FP) assays. Short peptides from the TDS region were unfolded in the solution, which we attributed to an imperfect hydrophobic repeat pattern, lacking leucine residues typical of the cores of dimeric coiled coils. Therefore, we made three mutations of core residues (A479, V486, K500) to leucine and templated folding using seven flanking residues on either side of the sequence from CC-Di (*Figure 2—figure supplement 2A*). Circular dichroism (CD) spectroscopy and analytical ultracentrifugation (AUC) confirmed that the resulting peptide (Eng-TDS) formed a helical dimer (*Figure 2—figure supplement 2B–E*). Subsequent FP assays showed that isolated KLC1-TPR interacted with Eng-TDS with low affinity (approx. *Kd* of 25 μM), but not with CC-Di alone (*Figure 2F*). We obtained similar results for KLC2-TPR with tighter binding to Eng-TDS (*Figure 2—figure supplement 2F*) (Kd = 2.7 μM), suggesting that this interaction is conserved between KLC paralogues. We note that these binding measurements are for an inter-molecular process, but in the context of the holoenzyme, the interaction is effectively intra-molecular because the TPR domain is tethered to the coiled-coil scaffold. Therefore, the effective local concentration is substantially higher, resulting in a high equilibrium fraction of bound complexes, consistent with our EM experiments. Prior work shows that constructs with deletions of the first helix of TPR1 of KLC2 result in unfolding of TPR1 (*Pernigo et al., 2013*; *Zhu et al., 2012*). Therefore, to test a key role for the TPR1 interface, we deleted the first helix of TPR1. This effectively abrogated any detectable binding (*Figure 2—figure supplement 2G*). Together, this modelling, electron microscopy analysis and biophysical experiments clearly demonstrate that the TDS provides a TPR domain docking site on the lambda particle.

## Cargo adaptor binding dislocates the TPR shoulder

To understand the consequences of cargo binding, we incorporated an activating W-/Y-acidic cargo adaptor ligand into the complex (KinTag) via fusion on a flexible linker at the carboxy-terminus of KLC; i.e., an intramolecular mimic of stable cargo binding, following a similar strategy used to solve the structure by X-ray crystallography (*Cross et al., 2021*; *Figure 3A*). In a wild-type background, this resulted in a shift in the SEC elution profile from the two peaks corresponding to open and closed conformations to a single peak at an intermediate position (*Figure 3B*), which negative stain EM analysis revealed consisted of a mixture of open and closed particles, but with no discrete intermediate state (*Figure 3—figure supplement 1*). From this, we conclude that the main effect of SLiM-TPR interaction is to increase the rate of exchange between open and closed conformations such that they are no longer resolved by SEC. Supporting this, additional incorporation of the closed-stabilising ElbowLock modification shifted the equilibrium to a position corresponding to the closed peak, demonstrating that the cargo-induced SEC shift can be suppressed by restricting conformational flexibility at the elbow, and therefore results from opening the complex (*Figure 3B*).

Analysis of the resulting ElbowLock-KinTag particles by NS-EM and 2D classification resulted in clear loss of the TPR shoulder (*Figure 3C–E*), showing that the intramolecular adaptor interaction

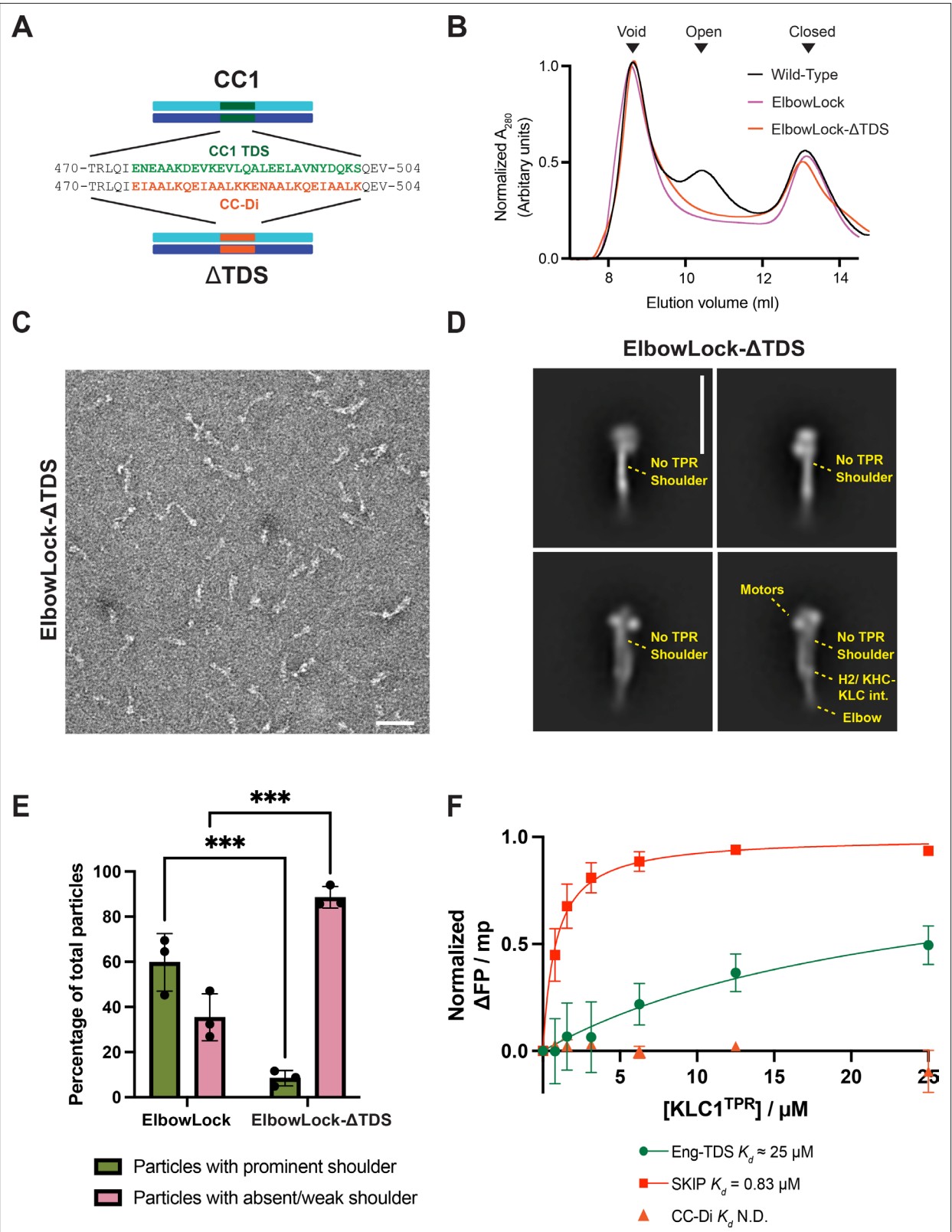

**Figure 2.** The CC1 TPR docking site (TDS) is required for the formation of the kinesin-1 shoulder. (**A**) Schematic showing strategy to delete the TDS in CC1 and replace it with an equivalent length of a de novo designed homodimeric coiled coil, CC-Di. (**B**) Size-exclusion chromatography (SEC) traces showing the elution profile of wild-type, ElbowLock, and ElbowLock-ΔTDS proteins. Constructs in the ElbowLock background elute exclusively in the peak associated with closed lambda particles. Representative of three independent experiments. (**C**) Representative negative stain (NS) electron

*Figure 2 continued on next page*

*Figure 2 continued*

micrographs (from three independent experiments) showing ElbowLock and ElbowLock-ΔTDS complexes. Scale bar is 50 nm (**D**) Selected 2D classes from NS-EM highlighting the presence of the shoulder in ElbowLock but not ElbowLock-ΔTDS complexes. The scale bar is 26 nm. Full sets of classes are provided in *Figure 1—figure supplement 4* (**E**) Quantification of number of particles in classes with and without a prominent shoulder of three independent experiments. *** indicates *p*<0.001 (**F**) Fluorescence polarisation binding assays show fluorescently labelled TDS and SKIP (W-acidic motif, positive control) peptides, but not the control peptide CC-Di, bind to isolated TPR domains. Error bars show S.E.M. from three replicates.

The online version of this article includes the following figure supplement(s) for figure 2:

**Figure supplement 1.** Alphafold3 models of ΔTDS kinesin-1 complexes.

**Figure supplement 2.** Biophysical characterisation of Eng-TPR docking site (TDS).

promotes TPR dissociation from the closed complex. To substantiate this further, we generated 3D models of ElbowLock, ElbowLock-ΔTDS and ElbowLock-KinTag complexes from the NS-EM data (*Figure 3—figure supplement 2*). For ElbowLock complexes, this resulted in classes with and without a prominent shoulder, in agreement with 2D classification. For ElbowLock-ΔTDS and ElbowLock-KinTag complexes, no prominent shoulder containing classes were observed. Moreover, these modifications appeared to result in a slight loosening of the whole coiled-coil assembly (*Figure 3F*); note the indentations between the densities of the two main coiled-coil regions, which we interpret as the start of a separation of these two domains that is restricted by the ElbowLock mutation. Consistent with a KinTag-induced inhibition of TDS-TPR association, inclusion of KinTag on the KLC-TPR carboxy-terminus effectively abrogated any detectable binding to Eng-TDS, or its direct competitor, the W-acidic motif of SKIP, in FP assays (*Figure 2—figure supplement 2H*). Together, these data support a model where adaptor-TPR interaction dislocates the TPR shoulder from the complex (*Figure 3G*).

## Cargo-mediated dislocation of the shoulder enhances motor domain accessibility

To determine how the local structural changes from adaptor binding and shoulder dislocation affected the dynamics of kinesin-1 complexes in solution, as directly and least invasively as possible, and without the risk of cross-linker artefacts, we recorded millisecond time-resolved hydrogen/deuterium-exchange mass spectrometry (HDX-MS) time courses, resolved at the peptide level (*Figure 4*, *Figure 4—figure supplements 1–4*; *Kish et al., 2023*; *Guo et al., 2022*; *Bai et al., 1993*; *Garcia et al., 2015*; *Fadgen, 2020*). Sequence coverage was good (overall 88%) with the exception of KHC-CC0 (neck coil) and the acidic-linker region that connects the KLC coiled-coil to the TPR domains where coverage was more limited.

To begin, we established baseline parameters by comparing the wild-type and DeltaElbow complexes. The latter stabilizes the fully extended conformation by promoting helical readthrough at the elbow and thus forces TPR undocking from CC1 (*Weijman et al., 2022*; *Cross et al., 2024*). The data for wild-type were subtracted from those of the DeltaElbow to generate difference plots (*Figure 4—figure supplements 3 and 4*), which were then mapped onto the X-ray crystal structures of the motor and TPR domains (*Figure 4*; *Zhu et al., 2012*; *Kozielski et al., 1997*).

First, as a general observation for the KHC chain, the largest differences in HDX were observed outside of its coiled-coil (CC) stalk. This is consistent with hydrogen-bonded α-helical structure and the known stabilities of coiled-coil assemblies in general that is protective against hydrogen exchange and may partially mask changes in solvent accessibility (*Woolfson, 2023*). Nonetheless, there were several small but statistically significant differences in labelling between the two constructs in the KHC CCs consistent with the opening of the DeltaElbow complex. These included deprotection/increased accessibility at the proposed motor-CC1 interface (*Tan et al., 2023*; *Carrington et al., 2024*), the TDS, and the KHC-KLC interface in CC3, running into the amino terminus of CC4 (*Figure 4—figure supplement 3A*). Interestingly, for the KLC chain in the open state, we observed more deprotection/increased accessibility in its CC and even protection at its amino terminus suggesting larger-scale activation-associated structural changes in this component of the kinesin complex (*Figure 4—figure supplement 4A*).

Second and strikingly, the DeltaElbow modification significantly increased HDX rates of the motor domains (*Figure 4A*). This is consistent with their liberation from the folded-back and locked conformation of the lambda particle (*Tan et al., 2023*). The increase in HDX was not tightly localised. Rather,

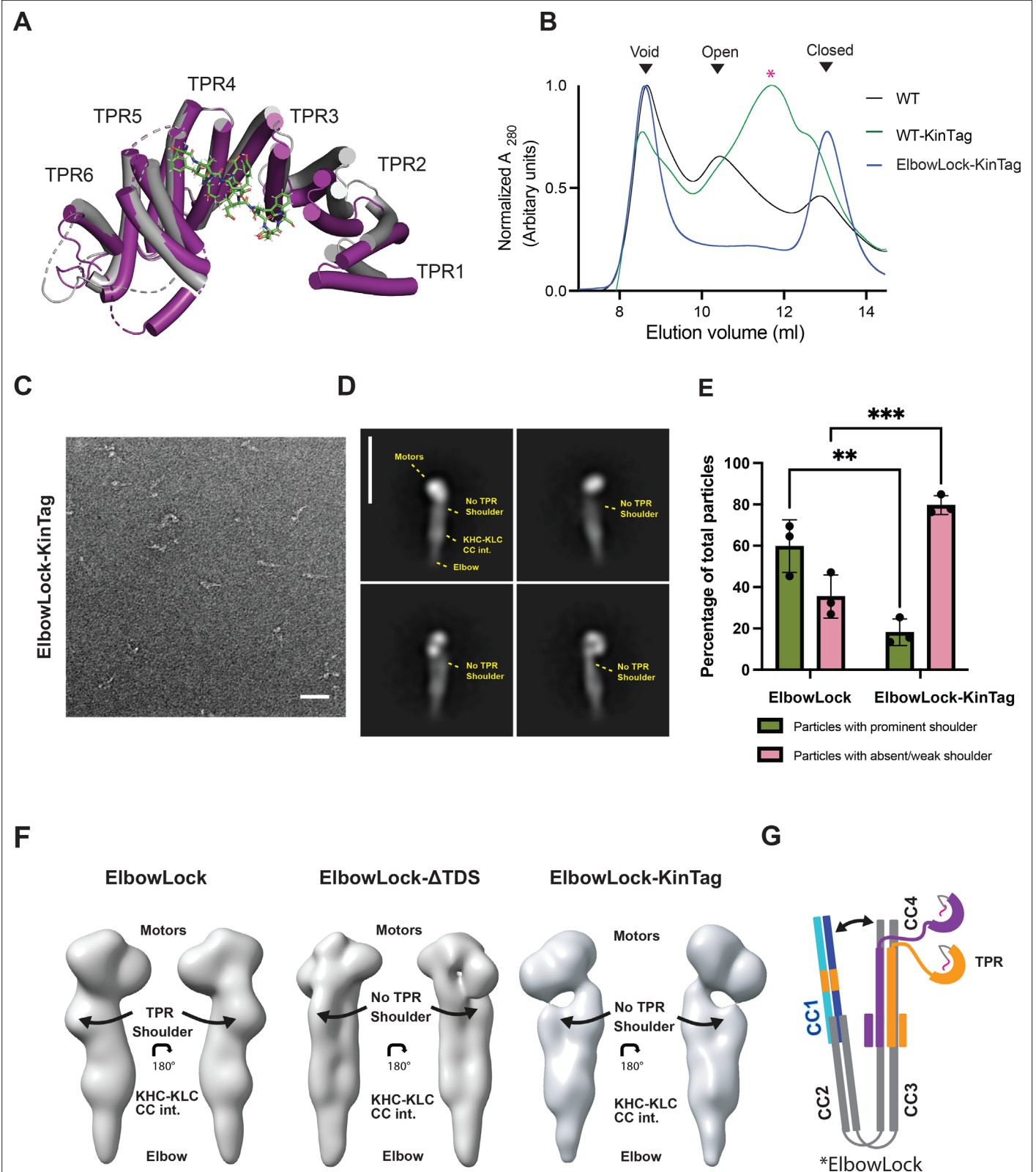

**Figure 3.** Adaptor binding to KLC-tetratricopeptide repeat (TPR) dislocates the kinesin-1 shoulder. (**A**) Comparison of X-ray crystal structures of ligand-free (grey, pdb: 3nf1) and KinTag (W/Y acidic)-ligand bound (purple, pdb: 6swu) TPR domains highlighting the ligand-binding site and ligand-induced change in TPR curvature. (**B**) Size-exclusion chromatography (SEC) traces showing elution profile of wild-type, ElbowLock, and ElbowLock-KinTag proteins. (**C**) Representative NS electron micrographs (from three independent experiments) showing ElbowLock and ElbowLock-KinTag

*Figure 3 continued on next page*

*Figure 3 continued*

complexes. Scale bar is 50 nm. (**D**) 2D classes from NS-EM highlighting the loss of the shoulder in ElbowLock-KinTag complexes. The scale bar is 26 nm. (**E**) Quantification of the percentage of particles in classes with and without a prominent shoulder from three independent experiments. ** indicates $p<0.001$ and *** indicates $p<0.001$ (**F**) 3D reconstructions from the ElbowLock, ElbowLock-ΔTDS, and ElbowLock-KinTag datasets show dislocation of the shoulder either upon deletion of the TDS or incorporation of KinTag. (**G**) Model showing dislocation of the shoulder induced upon KinTag binding.

The online version of this article includes the following figure supplement(s) for figure 3:

**Figure supplement 1.** Representative negative stain electron microscopy (NS-EM) electron micrograph showing wild-type KinTag particles.

**Figure supplement 2.** Electron microscopy (EM) processing workflow for 3D reconstructions.

the motor domain as a whole was significantly deprotected. This is consistent with enhanced solvent accessibility and increased local subdomain dynamics throughout the motors in response to opening of the elbow. In addition, opening the complex resulted in significant deprotection/increased solvent accessibility of TPR1. This is consistent with its role in docking the TPR onto the CC assembly and its forced undocking by helical readthrough in the DeltaElbow variant. We also observed additional deprotection of TPR6, extending through to the C-terminal sequence that follows the TPR, indicating possible occlusion of these features in the inhibited state.

Next, we compared the ElbowLock (stabilised closed) and wild-type complexes. Opposite to DeltaElbow, both the motor domains and the TPR1 sites were more protected/less accessible in the ElbowLock variant (*Figure 4B*, *Figure 4—figure supplements 3B and 4B*). There was also protection of the C-terminal tail of KHC. Thus, shifting the open-closed equilibrium to either of its extremes enables visualisation of the accessibility of key interfaces in the complex that maintain autoinhibition, and support our findings emerging from AF3 modelling, negative-stain EM, and biophysical analyses.

Finally, we compared the ElbowLock-KinTag and ElbowLock complexes to examine the consequences of adaptor binding in the closed state (*Figure 4C*, *Figure 4—figure supplements 3C and 4C*). As expected, the inclusion of KinTag significantly decreased HDX in, and therefore stabilised, the central TPR repeats at its structurally characterised binding site (*Cross et al., 2021*). This was juxtaposed with the striking deprotection of TPR1, which is consistent with ligand-induced TPR domain undocking from the lambda particle at this epitope (*Figure 3*). The motor domains were also deprotected. Changes on the KHC CC were again much lower in magnitude, but we did note statistically significant deprotection at the predicted motor-CC1 docking site and C-terminal tail, suggesting that TPR-adaptor binding triggers release of the motors from their folded back, locked conformation.

Together, these data provide direct in-solution measurements revealing both predicted and unanticipated changes in solvent accessibility associated with the opening and closing of the complex that demonstrate allosteric coupling between adaptor-induced dislocation of the TPR shoulder and increased motor domain accessibility.

## Opening the kinesin-1 complex enhances its association with MAP7

Thus far, our findings support a model where cargo-binding destabilises the compact conformation and so may unmask protein-protein interaction interfaces that facilitate subsequent steps in the activation pathway. The microtubule-associated protein MAP7 is an essential kinesin-1 cofactor that promotes binding to microtubules and activation. Its binding site has been biochemically mapped to a short region of CC1 that overlaps the TDS (*Barlan et al., 2013*; *Ferro et al., 2022*; *Hooikaas et al., 2019*; *Chaudhary et al., 2019*; *Metzger et al., 2012*; *Tymanskyj et al., 2018*; *Berisha et al., 2025*; *Monroy et al., 2018*). Indeed, AlphaFold3 predictions suggest that although the TPRs and MAP7 may bind to CC1 via distinct mechanisms (near perpendicular TPR helix-turn-helix motif vs parallel single MAP7 alpha helix), their binding sites directly overlap, and as such, they may compete for KHC binding (*Figure 5A*). One prediction of this model is that the opening of the complex at the Elbow to force TPR undocking would enhance binding to MAP7. To test this, we immunoprecipitated GFP-tagged MAP7 from cells expressing wild-type, DeltaElbow and ElbowLock kinesin-1 complexes. This resulted in a roughly fivefold increase in kinesin-1 associated with MAP7 in the DeltaElbow background, compared to wild-type or ElbowLock constructs (*Figure 5B and C*). We also noted slightly elevated expression of ElbowLock complexes and slightly lower expression of DeltaElbow complexes, suggesting that opening/closing of the complex could impact kinesin-1 turnover. Similarly, in immunofluorescence experiments, the DeltaElbow modification resulted in a striking increase in kinesin-1

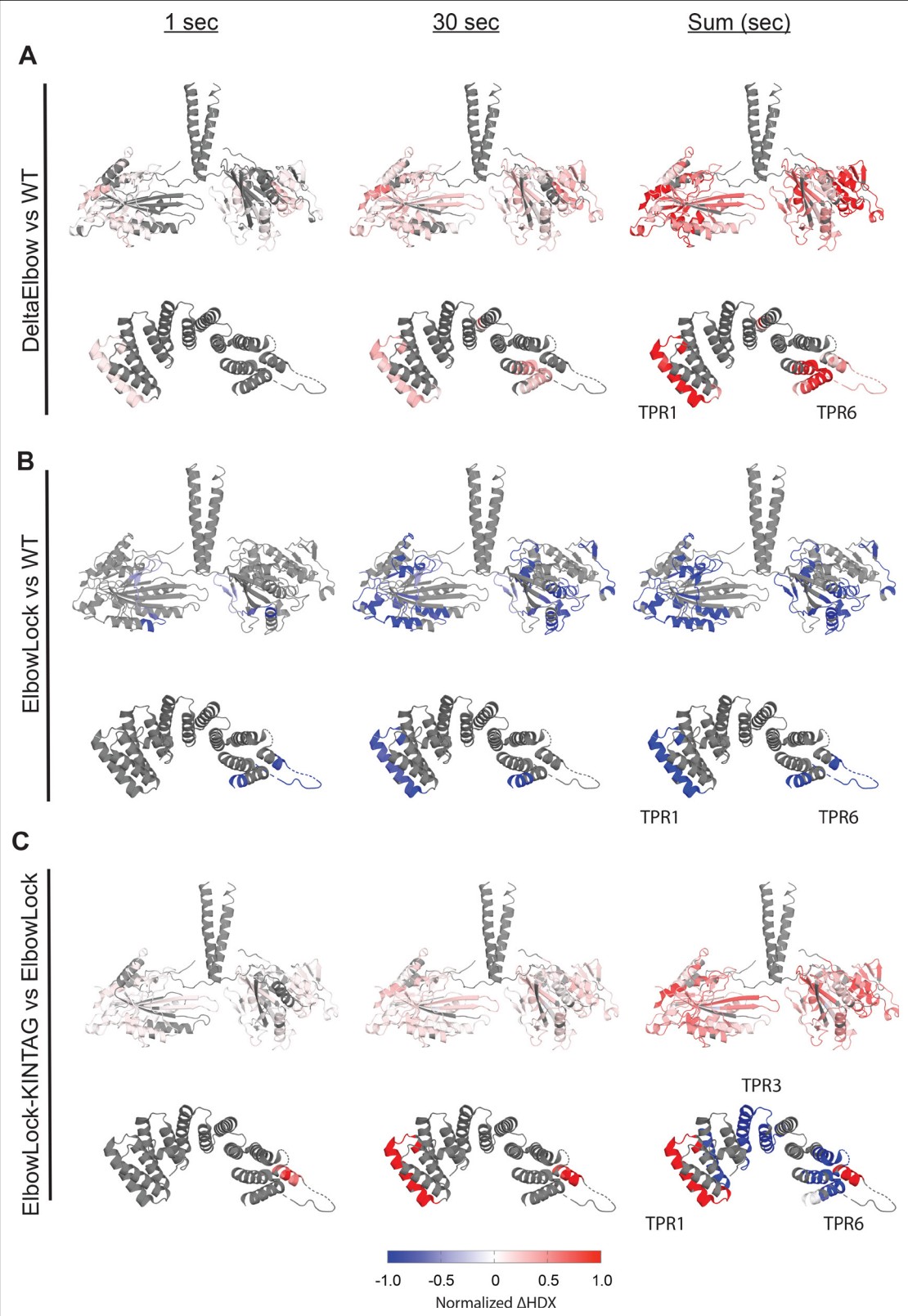

**Figure 4.** Adaptor binding to the kinesin light chain (KLC) tetratricopeptide repeat (TPR) domain promotes motor accessibility. Summaries of comparative hydrogen/deuterium-exchange mass spectrometry (HDX-MS) analysis of wild-type, DeltaElbow, ElbowLock, and ElbowLock-Kintag complexes. At each time point, the rates of exchange were subtracted as indicated (X minus Y) and filtered using a hybrid significance test global significance and Welch's t-test, then transposed onto X-ray crystal structures of the motor domains (pdb: 3kin) and TPR domain (pdb: 3nf1). Grey

*Figure 4 continued on next page*

*Figure 4 continued*

indicates non-significant differences or limited peptide coverage. (**A**) Comparison of deltaElbow (**X**) against wild-type (**Y**). (**B**) ElbowLock (**X**) against wild-type (**Y**). (**C**) ElbowLock-Kintag (**X**) against ElbowLock (**Y**). Relative protection (blue) indicates less solvent exposure or increased H-bonding for X, while deprotection (red) indicates more solvent exposure or H-bond loss for X.

The online version of this article includes the following source data and figure supplement(s) for figure 4:

**Source data 1.** Hydrogen/deuterium-exchange mass spectrometry (HDX-MS) summary table.

**Figure supplement 1.** Schematic representation of the hydrogen/deuterium-exchange (HDX) experiment.

**Figure supplement 2.** Coverage map of mass spectral assignment.

**Figure supplement 3.** Difference plots showing total hydrogen/deuterium-exchange (HDX) across timepoints comparing heavy chain of DeltaElbow and Elbowlock to wild-type and ElbowLock-KinTag to ElbowLock.

**Figure supplement 4.** Difference plots showing total hydrogen/deuterium-exchange (HDX) across all timepoints comparing light chain of DeltaElbow and Elbowlock to wild-type and ElbowLock-KinTag to ElbowLock.

associated with MAP7-positive microtubules and pronounced microtubule bundling (*Figure 5D*). Together, these data reveal a coupling between kinesin-1 conformational state and its capacity to associate with the microtubule-associated co-factor MAP7.

## Discussion

Over the past two decades, studies have identified a key role for SLiM-containing adaptor proteins in recruiting and activating kinesin-1 by binding to the KLC-TPR domains (*Cross and Dodding, 2019*; *Verhey et al., 2001*; *Pernigo et al., 2013*; *Cross et al., 2021*; *Dodding et al., 2011*; *Chiba et al., 2022*; *Kawano et al., 2012*; *Araki et al., 2007*; *Konecna et al., 2006*; *Pernigo et al., 2018*). Recent work has revealed the low-resolution architecture of the autoinhibited lambda state, including the TPR shoulder resting on the coiled-coil body (*Tan et al., 2023*; *Weijman et al., 2022*; *Carrington et al., 2024*). Increasing evidence suggests that a balanced equilibrium between inhibited and active states ensures responsiveness to stimuli (*Cross et al., 2024*; *Chiba et al., 2022*; *Smith et al., 2024*). The challenge is to understand how adaptor binding to the TPR drives the transition from the closed, inhibited state to the open, active state; intrinsic conformational dynamics and unitarily low-affinity, cooperative, motor-cargo interactions obscure key intermediates in this pathway. To address this here, we have used de novo protein design and protein engineering to stabilise these intermediates along-side biophysical, biochemical, and electron microscopy measurements of the complete heterotetra-metric holoenzyme. Our results lead us to propose a model where adaptor binding dislocates the TPR shoulder, destabilizing the autoinhibited state, enhancing motor domain accessibility, and promoting the transition from the inactive closed state to the open forms (*Figure 6*).

Integration of the present findings with prior work suggests a unified model for the role of TPR binding cargo adaptors in motor activation. We reveal that TPR1, the first TPR repeat, forms a key interface for TPR domain-CC association. TPR1 also binds the adaptor protein JIP3 via its coiled-coil leucine zipper, resembling TDS interaction predictions. However, a detailed evolutionary analysis in the same study suggests that TPR1 plays a distinct role beyond cargo binding (*Cockburn et al., 2018*). Moreover, a comparison of high-resolution X-ray crystal structures of KLC-TPR domains, with and without SLiM cargo, demonstrates structural plasticity in TPR1, which may enable ligand binding to modulate TPR1 orientation alongside larger scale conformational changes resulting from an increase in TPR domain curvature in response to SLiM binding (*Yip et al., 2016*; *Pernigo et al., 2013*; *Pernigo et al., 2018*; *Nguyen et al., 2017*). We propose that SLiM ligand binding to the TPR groove drives conformational changes that reduce TPR1 affinity for the TDS to dislocate the shoulder; JIP3 may compete for the same site on TPR1 to the same effect. Our previous work has shown that SLiM-TPR interaction gates access to important cargo and microtubule binding sites on CC4, that is predicted to be closely juxtaposed to CC1 in models of the inhibited state (*Yip et al., 2016*; *Randall et al., 2017*; *Sanger et al., 2017*). Our computational predictions do not indicate a direct interaction between KLC TPR and CC4, although they are predicted to be in proximity. Whilst we do not exclude this possibility of a direct interaction, we favour a model where removal of steric blocking effects from the bulky TPRs and docked motors on CC1, as well as opening of the coiled-coil assembly, enables subsequent steps, such as cargo binding to additional sites on CC4.

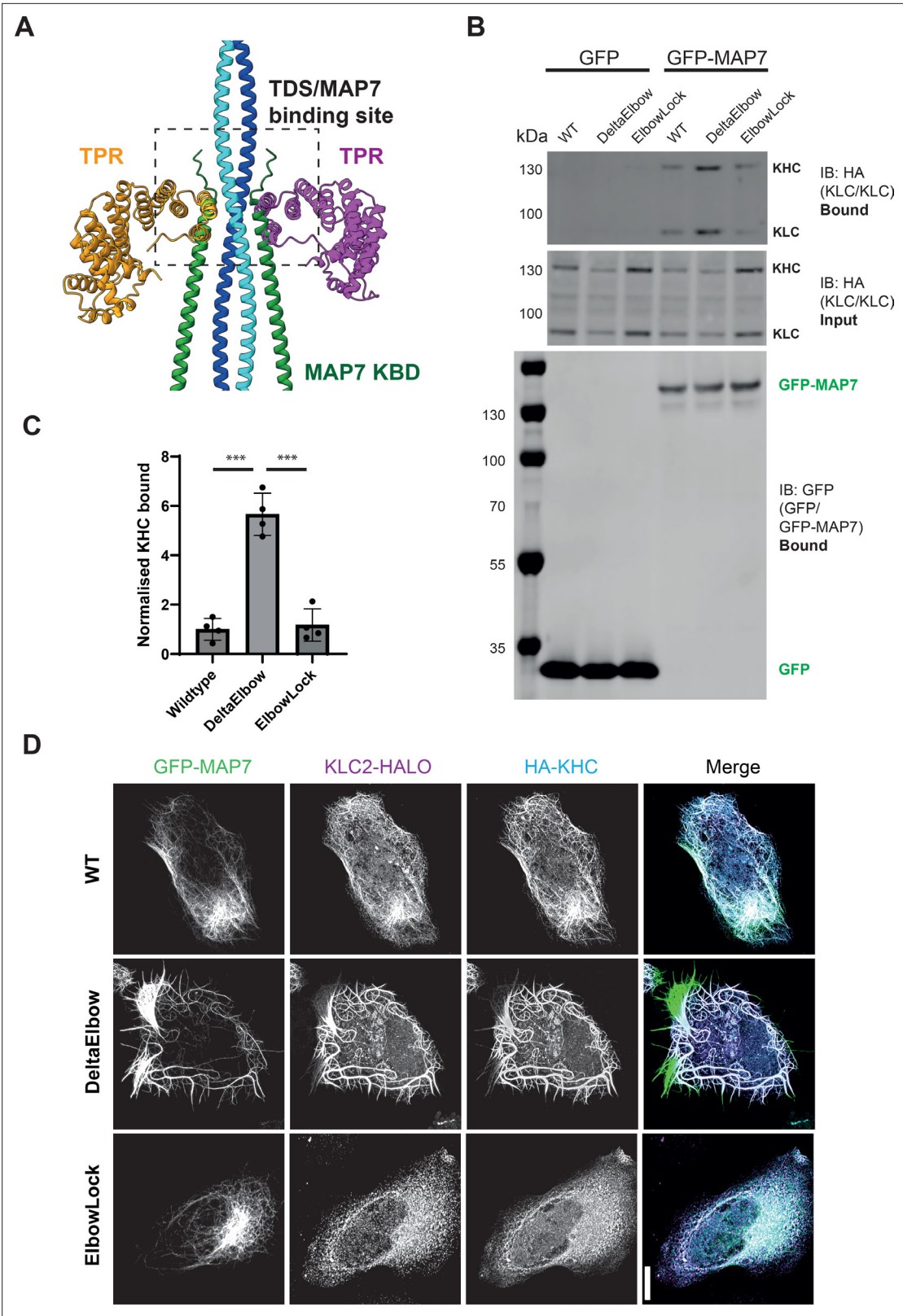

**Figure 5.** Opening the kinesin-1 complex promotes its association with MAP7. (**A**) Overlay of two Alphafold3 models of a tetrameric CC1/TPR assembly and a tetrameric CC1/MAP7-kinesin binding domain assembly, aligned on CC1. For clarity, only one CC1 model is shown. (**B**) GFP-MAP7 immunoprecipitation experiments showing enhanced association of kinesin heavy chain (KHC) and kinesin light chain (KLC) in the DeltaElbow background. Cells were transfected with the indicated constructs GFP/GFP-MAP7, HA-KLC2, HA-KIF5C (WT, ElbowLock, DeltaElbow). Complexes were

*Figure 5 continued on next page*

*Figure 5 continued*

immunoprecipitated using GFP-TRAP beads, and input and bound samples were analysed using western blotting with anti-GFP and anti-HA antibodies. (**C**) Quantification of b. from 4 independent experiments. Error bars show S.E.M. ***$p<0.001$ using one-way ANOVA with Tukey's multiple comparison test. (**D**) Immunofluorescence images of HeLa cells transfected with GFP-MAP7, HA-KHC, and KLC2-Halo (TMR-labelled) showing enhanced association of kinesin-1 with GFP-MAP7 positive microtubules in the DeltaElbow background. Images are representative of three independent experiments. Scale bar is 20 μm.

The online version of this article includes the following source data for figure 5:

**Source data 1.** Uncropped western blots relating to *Figure 5B*.

**Source data 2.** Raw TIFF files for western blots relating to *Figure 5B*.

The TDS is located within a biochemically defined binding site for the essential kinesin-1 cofactor MAP7 (*Barlan et al., 2013*; *Ferro et al., 2022*; *Hooikaas et al., 2019*; *Chaudhary et al., 2019*; *Metzger et al., 2012*; *Tymanskyj et al., 2018*; *Berisha et al., 2025*; *Monroy et al., 2018*). Recent studies show clear cooperativity between MAP7 and W-acidic SLiM adaptors in motor activation (*Chiba et al., 2022*; *Sahabandu et al., 2025*). This work provides a compelling explanation for this co-operativity: shoulder dislocation would be predicted to destabilise autoinhibition and expose the MAP7 binding site, facilitating subsequent activation steps, such as enhanced microtubule recruitment and transition to and/or maintenance of the fully extended state (*Figure 6*). Consistent with this, our models suggest that CC1 binding to MAP7 or KLC TPR would be mutually exclusive *if* the symmetrical binding sites on CC1 were both occupied (2xMAP7 or 2xKLC-TPR). However, it is not clear at this point whether this is the case. In the absence of nanobody-mediated TPR stabilisation, negative stain EM hints at an asymmetric TPR conformation, and we remain open to the possibility that only one TPR is bound at one time and that they may exchange. This would, in principle, leave one CC1 binding site free to interact with MAP7 and one TPR domain free to make first contact with cargo.

It will be important to understand this dynamic interplay and to sequence these events. A cargo-initiated opening up of the complex is consistent with a model for MAP7 function where it facilitates tethered diffusion to bypass obstacles on the microtubule and enhances kinesin-1 processivity (*Ferro et al., 2022*; *Hooikaas et al., 2019*), and activation without cargo recognition would seem counter-productive. However, recently, MAP7 binding to KHC was shown to promote KLC binding to W-acidic and Y-acidic cargo (*Sahabandu et al., 2025*). Here, we have used a high-affinity ligand to mimic stable cargo attachment, but natural motifs typically bind with low affinity and act cooperatively with additional protein-protein and protein-lipid interactions (*Antón et al., 2021*; *Pernigo et al., 2013*; *Dodding et al., 2011*; *Sanger et al., 2017*; *Blasius et al., 2007*). Therefore, one might also consider a two-way switch model where MAP7 binding also facilitates shoulder dislocation to modulate cargo binding properties of the KLC TPR domains. In either case, this points to a critical interplay between regulator and cargo, converging on CC1, establishing a new target for intervention in kinesin-1 mediated transport processes, and design of new reagents to target this site will likely be critical to dissect its dynamic functions.

In summary, our findings demonstrate that kinesin-1 autoinhibition and initiation of activation by SLiM-cargo pivots on the association and dislocation of its TPR shoulder from the coiled-coil scaffold. One important implication of these findings is that kinesin-1 is unlikely to be stably inhibited when bound to cargo but instead may be primed for further regulatory inputs that control its activity. This may be important for the coordination of bidirectional transport in multi-motor/adaptor systems.

## Materials availability statement

Plasmid DNAs used in this study are available upon request to mark.dodding@bristol.ac.uk.

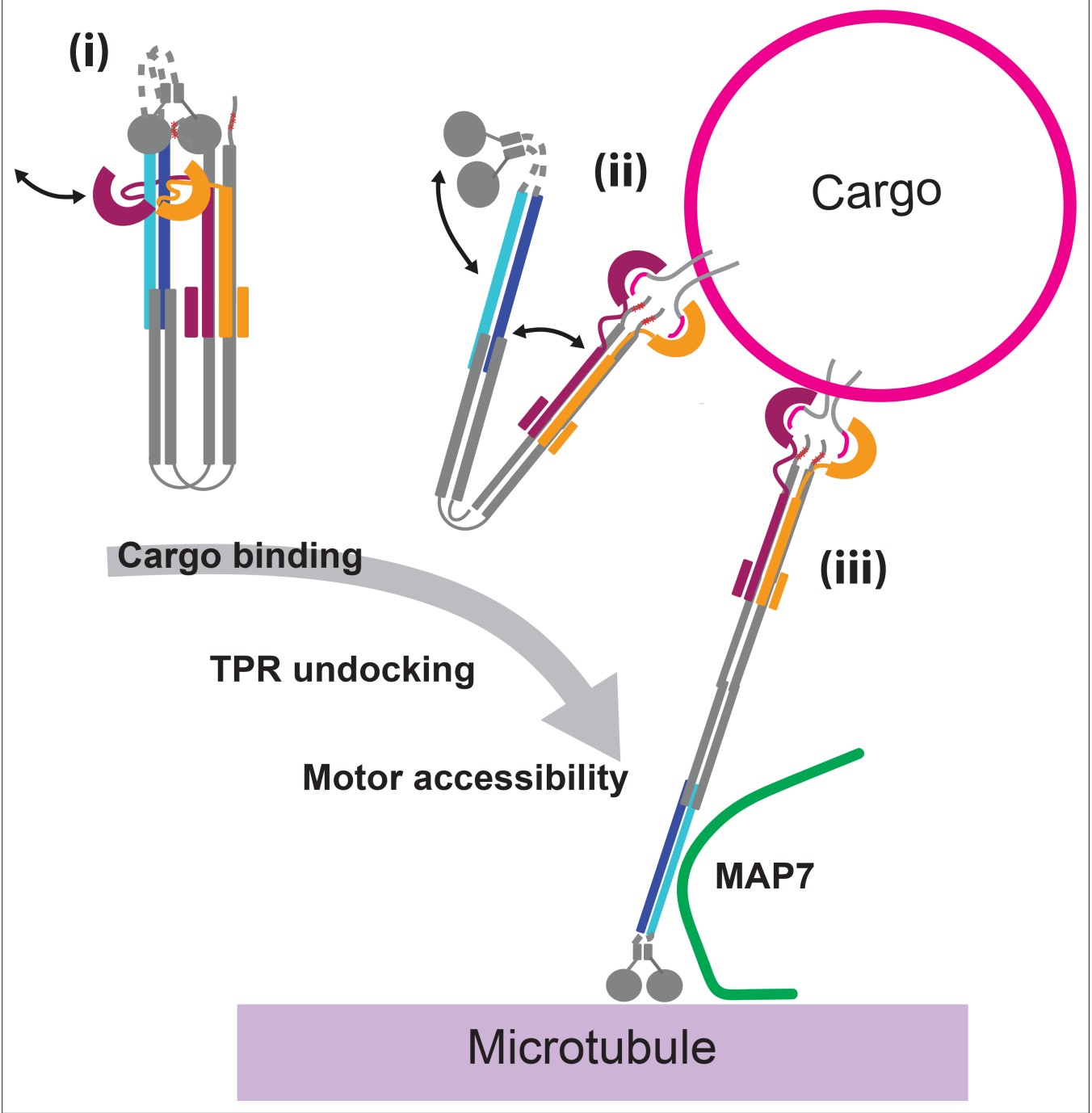

**Figure 6.** Model for cargo-mediated initiation of kinesin-1 activation. In the autoinhibited state, a tetratricopeptide repeat (TPR) domain(s) are docked at the TPR docking site (TDS) forming the shoulder. (i). Binding to the cargo adaptor dislocates the TPR shoulder and promotes motor domain accessibility, and initiates separation of the coiled-coil domains (ii). This allosteric coupling between cargo and motor would facilitate recruitment to microtubules (supported by MAP7) and subsequent transport (iii).

# Materials and methods

**Key resources table**

| Reagent type (species) or resource | Designation | Source or reference | Identifiers | Additional information |
|---|---|---|---|---|
| Recombinant DNA reagent | pMW-His-KIF5C | *Weijman et al., 2022* | | pMW vector, N-terminal 6×His tag, for bacterial expression. |
| Recombinant DNA reagent | pET28-KLC1 | *Weijman et al., 2022* | | pET28, His tag removed, for bacterial expression. |
| Recombinant DNA reagent | pMW-His-KIF5C (ElbowLock) | *Cross et al., 2024* | | KIF5C, elbow modification, for bacterial expression. |
| Recombinant DNA reagent | pMW-His-KIF5C (DeltaElbow) | *Weijman et al., 2022* | | KIF5C, elbow modification, for bacterial expression. |
| Recombinant DNA reagent | pMW-His-KIF5C-ΔTDS | This paper | | KIF5C, CC1 modification, for bacterial expression. |
| Recombinant DNA reagent | KLC1-KinTag | This paper | | pET28, His tag removed, for bacterial expression, C-terminal KinTag fusion via $(TGS)_{10}$ linker |
| Recombinant DNA reagent | KLC1 TPR domain | *Pernigo et al., 2013* | | pET28, His tagged KLC1 TPR domain |
| Recombinant DNA reagent | KLC2 TPR domain | *Pernigo et al., 2013* | | pET28, His tagged KLC2 TPR domain |
| Recombinant DNA reagent | KLC2 TPR domain delta Helix 1 | *Pernigo et al., 2013* | | pET28, His tagged KLC2 TPR domain, helix 1 removed |
| Recombinant DNA reagent | Anti-TPR nanobody | *Pernigo et al., 2018* | | Synthesised (GenScript), pelB leader, C-terminal His tag |
| Recombinant DNA reagent | GFP-MAP7 | *Metzger et al., 2012* | Addgene plasmid #46076 | Mammalian expression |
| Recombinant DNA reagent | HA-KLC2 | *Sanger et al., 2017* | | Mammalian expression |
| Recombinant DNA reagent | HA-KIF5C | *Sanger et al., 2017* | | Mammalian expression |
| Recombinant DNA reagent | KLC2-Halo | *Yip et al., 2016* | | Mammalian expression |
| Recombinant DNA reagent | HA-KIF5C DeltaElbow | *Weijman et al., 2022* | | Mammalian expression |
| Recombinant DNA reagent | HA-KIF5C ElbowLock | *Cross et al., 2024* | | Mammalian expression |

## AlphaFold3 modelling

All of the models shown in the manuscript were predicted using the AlphaFold3 server (*Abramson et al., 2024*). The algorithm was asked to model a heterotetramer composed of KIF5C (NP_001101200.1) and KLC1A (NP_032476.2) with varying input sequences described in the figure legends. Predicted aligned error statistics are as output by the software with manual annotations to highlight important features. For the models presented in *Figure 5*, the algorithm was asked to model a heterotetramer of KIF5C CC1 (S410-G558) with the kinesin binding helix of MAP7 (NP_001185564.1, P467-K610), or KLC1 TPR domain (NP_032476.2, G205-K495). Similar models and predicted binding sites were obtained for heterotrimers (2xCC1, 1xMAP7/TPR). Output models were visualised/aligned and prepared for publication using UCSF Chimera (*Pettersen et al., 2004*).

## Constructs for protein expression

A 2-plasmid system to express heterotetrameric kinesin-1 in *E. coli* cells has been described previously (*Weijman et al., 2022*; *Cross et al., 2024*) and constructs used in this study were derived from these. Wild-type kinesin heavy chain was KIF5C expressed in the pMW bacterial expression vector with an N-terminal 6 X His tag. KLC1 was expressed in pET28a with the His tag removed by site-directed mutagenesis. ElbowLock was originally designed by deleting five amino acids from the elbow loop (E685-L690) (*Cross et al., 2024*), and ElbowLock-ΔTDS was prepared by additionally replacing E475-S501 in CC1 with 4-heptads of CC-Di through subcloning of a synthetic gene fragment. KLC1-KinTag

construct was prepared by fusing the KinTag peptide via a (Thr-Gly-Ser)$_{10}$-Gly flexible linker at the C-terminal of KLC1 (*Cross et al., 2021*). The TPR domain of KLC1 (A211-A495) and KLC2 (A196-K480, delta helix 1 P219-K480) were cloned in the bacterial expression vector pET28His-Thrombin and have been described previously (*Yip et al., 2016*; *Pernigo et al., 2013*). Anti-TPR nanobody (sequence from pdb: 6fv0)(*Pernigo et al., 2018*) was codon-optimised and synthesised and cloned into NcoI and XhoI site of pet22b with N-terminal pelB sequence and C-terminal 6 X His tag, by Genscript Inc The identity of all plasmids was confirmed by DNA sequencing.

## Protein expression and purification

All kinesin-1 constructs were expressed in *E. coli* BL21(DE3) cells as a heterotetramer adapting the two-plasmid system as described previously (*Weijman et al., 2022*; *Cross et al., 2024*). Briefly, expression plasmids encoding individual KIF5C constructs and KLC1 constructs were transformed into *E. coli* BL21(DE3). Positive clones were selected based on a double antibiotic selection marker. Cells were cultured in LB medium (Miller) supplemented with ampicillin (50 µg/ml) and kanamycin (50 µg/ml) in a shaking incubator under conditions of 37 °C/180 rpm. When the optical density (OD 600) measured at 600 nm reached 0.6, the temperature was reduced to 18 °C. Protein expression was induced by addition of 0.2 µM IPTG. Cultures were incubated overnight with shaking at 18 °C. Cells were harvested by centrifugation at 5000 g for 15 min at 4 °C. Cells were resuspended in Buffer A (40 mM Hepes (pH 7.4), 500 mM NaCl, 40 mM imidazole, 5% glycerol, and 5 mM β-mercaptoethanol), supplemented with protease inhibitors. Lysis was carried out by sonication (70% amplitude, 5 s on/15 s off) for 7.5 min on ice. Lysates were centrifuged at 40,000 g for 45 min at 4 °C to obtain supernatant and filtered through a 0.45 µm membrane. The supernatant was loaded onto a His-Trap (1 ml) column equilibrated with Buffer A. The column was washed with at least 50 CV of Buffer A with 60 mM Imidazole. Bound proteins were eluted using a gradient of Buffer A with increasing concentrations up to 500 mM of Imidazole. Eluted protein fractions were analysed on SDS-PAGE, then concentrated using a 30,000 Da MWCO ultrafiltration device up to 500 µl and immediately loaded on a size exclusion chromatography (SEC) column. SEC was carried out using a Superose 6 10/300 column pre-equilibrated with 20 mM Hepes (pH 7.4), 150 mM NaCl, 1 mM MgCl2, 0.1 mM ADP, and 0.5 mM TCEP. Eluted protein fractions were analysed using SDS-PAGE. Appropriate fractions were aliquoted and stored at –80 °C freezer and used further for downstream applications.

TPR constructs were expressed in *E. coli* BL21(DE3) and affinity-purified in buffer (25 mM Hepes pH 7.4, 500 mM NaCl, 5 mM β-mercaptoethanol) over a gradient of 20 mM to 500 mM imidazole on a His-Trap (1 ml) column (*Yip et al., 2016*; *Pernigo et al., 2013*). SEC was carried out using a HiLoad 16/600 Superdex 75 prep grade column (GE Healthcare) equilibrated with 25 mM HEPES (pH 7.4), 500 mM NaCl, 5 mM β-mercaptoethanol. Eluted fractions were analysed on SDS-PAGE.

Nanobody was expressed in *E. coli* BL21(DE3) and isolated from periplasmic space in 0.2 M Tris pH 8, 0.5 M EDTA, 0.5 M Sucrose, and purified on His-Trap (1 ml) column. SEC was carried out using a Superose 6 10/300 column pre-equilibrated with PBS. Eluted fractions were concentrated and mixed in 10-fold excess with SEC-purified ElbowLock (5.56 µM: 66 µM). Eluted complexes were analysed on SDS-PAGE.

## Negative stain electron microscopy

Protein aliquots (50 µl) were thawed from –80 °C storage and cross-linked with 0.6 µl of 50 mM BS3 (final concentration 0.59 mM) at room temperature for 30 min. After cross-linking, the protein sample was placed on ice, and serial dilutions were prepared to achieve optimal single-particle distribution on grids. Formvar/carbon-coated 300-mesh copper grids were glow-discharged for 30 s at 20 mA. A 5 µl aliquot of the protein sample was applied to the grid and incubated for 1 min before blotting with filter paper. The grid was subsequently stained by sequentially picking and blotting into three 5 µl drops of 3% uranyl acetate. The first drop served as a quick rinse, the second was left on the grid for 2 min, and the third also served as a quick rinse. The prepared grids were air-dried and subsequently imaged on a Talos L120C transmission electron microscope operated at 120 kV, equipped with a 4k × 4k Ceta CMOS camera. Micrographs were acquired at a magnification of ×57,000, corresponding to a pixel size of 2.48 Å/pixel. Datasets of approximately 500 micrographs were acquired using EPU automated collection software for all samples. The total electron dose was 52 e–/Å², and micrographs were recorded with defocus values ranging from –1.5 µm to –2.5 µm.

## Negative stain electron microscopy image processing

Micrographs were analysed using CryoSPARC (*Punjani and Fleet, 2022*). After importing the micrographs, CTF correction was performed using the Patch CTF tool. Approximately 1,000 particles were manually picked and subjected to 2D classification to generate templates for automated particle picking. Using these templates, particles were autopicked, extracted (63.4 nm box size), and Fourier-cropped to a pixel size of 4.96 Å/pixel. Extracted particles were 2D classified into 200 classes. A cleanup process was conducted by discarding particles that did not resemble kinesins through iterative rounds of curation and 2D classification. The number of particles in classes showing full 'side views' of the complex were counted and distinguished as particles with or without a prominent shoulder. The data quantified in this way was statistically analysed using two-way ANOVA (using Sidak's multiple comparison test). For figure presentation only, and to allow for intuitive comparison to AlphaFold models, final particle sets were reclassified with the CryoSPARC 'align filament classes vertically' option ticked. This resulted in no obvious morphological differences, but produced classes aligned on the long axis of the complex. For 3D reconstruction, cleaned particle sets were used to generate four ab initio models in CryoSPARC. A heterogeneous refinement of ab initio models was performed, followed by homogeneous refinement, with alignment resolution restricted to 20 Å. Resulting models were low-pass filtered to 40 Å resolution and visualised using ChimeraX (*Pettersen et al., 2004*).

## Cryo-electron microscopy and image processing

Purified cross-linked ElbowLock complexes (5 µl at 3 mg/ml) were applied to glow-discharged Quantifoil R1.2/1.3 200-mesh Cu grids. The grids were blotted for 2 s at 4 °C and 100% humidity, then flash-frozen in 37% ethane/propane mix kept at approximately −195 °C using an FEI Vitrobot Mark IV (Thermo Fisher). Images were acquired on a Talos Artica microscope (FEI) operating at 200 kV with a Gatan K2 Summit direct detector. Automatic image collection was performed using EPU software (Thermo Fisher). A total of 3943 micrographs were captured with a total dose of 58 e/Å², dose-fractionated into 64 movie frames at a defocus ranging from −1 µm to −2.5 µm. Images were recorded at ×130,000 nominal magnification, resulting in a pixel size of 1.05 Å per pixel. Data processing was carried out in CryoSPARC (*Punjani and Fleet, 2022*). Motion correction and CTF estimation were performed using Patch Motion Correction and Patch CTF, respectively. Particles (focusing on full-length side views) were manually picked from selected micrographs to create templates for automated particle picking. The autopicked particles were extracted and binned to 4.2 Å/pixel with a box size of 128 pixels/53.76 nm Extracted particles underwent multiple rounds of 2D classification to produce the class averages shown in *Figure 1*. For comparison to the AF3 model, simulated density was generated using the molmap command in ChimeraX (*Pettersen et al., 2004*) filtering to 15 Å. and projections were generated/selected automatically using the Reference Based Auto Selected 2D function in CryoSPARC.

## Hydrogen-deuterium exchange mass spectrometry

Hydrogen-deuterium exchange (HDX) was conducted using 'ms2min,' a fully automated, millisecond-resolution HDX labelling online quench-flow device (Applied Photophysics Ltd), which was connected to a Waters HDX manager. In the labelling experiments, 14 µL (10 µM) of DeltaElbow, wild-type, Elbow-Lock, and ElbowLock-KinTag was introduced into the labelling mixer separately. The HDX reaction was initiated by diluting the sample 20-fold with a labelling buffer at 20°C. This buffer consisted of 20 mM HEPES, 150 mM NaCl, 1 mM MgCl2, 0.1 mM ADP, 0.5 mM TCEP in D2O, adjusted to a pH of 7.40 at 20 °C. Samples were labelled at seven different time points (0.3, 0.5, 1, 3, 10, 30, 300) in triplicate. The HDX reaction was immediately quenched by mixing 1:1 with quench buffer (8 M urea, pH 2.55, at 0 C), and the sample was digested online using a Waters Enzymate pepsin column. The resulting peptides were trapped on a 2.1×5 mm VanGuard ACQUITY BEH C18 column (Waters) for 3 min at a flow rate of 155 µL/min and then separated on a 1×100 mm ACQUITY BEH C18 column (1.7 µm particle size) using a 7 min linear gradient of 5–40% acetonitrile with 0.1% formic acid. Peptides were analysed using a Synapt G2-Si mass spectrometer (Waters) in HDMSE mode across a mass range of 50–2000 m/z. Instrument settings were as follows: capillary voltage of 3.0 kV, cone voltage of 50 V, trap collision energy of 4 V, traveling wave ion mobility with a speed of 475 m/s, wave amplitude of 36.5 V, and nitrogen pressure of 2.75 mbar. Low-energy scans applied a transfer collision energy of 4 V, while high-energy scans utilised four distinct collision energy ramps between 15 and 55 V.

ProteinLynx Global Server (PLGS 2.5.1, Waters) was employed to analyse MSE reference data and to identify all detectable peptic peptides. The raw data files were processed, and isotopic distributions assigned using DynamX 3.0 (Waters, USA). The mean values and standard deviations (SD) of these replicates were calculated to assess the significance of changes, with a global significance threshold applied and a T-test (*Hageman and Weis, 2019*). The peptide level data were also flattened per amino acid (*Seetaloo et al., 2022*; *Keppel and Weis, 2015*). The Hydrobot software package was used for post-processing analysis (*Kish and Phillips, 2026*), and visualisations were generated using PyMOL (Schrödinger, Inc).

### Peptide synthesis

Standard Fmoc solid-phase peptide synthesis was performed on a 0.1 mM scale using CEM (Buckingham, UK) Liberty Blue automated peptide synthesis apparatus with inline UV monitoring. Activation was achieved with DIC/Oxyma. Fmoc deprotection was performed with 20% v/v morpholine/DMF. Double couplings were used for β-branched residues and the subsequent amino acid. Peptides were synthesised from C to N terminus as the C-terminal amide on Rink amide resin. For fluorescently labelled peptides, TAMRA (0.1 mM, 2 eq.), HATU (0.095 mM, 1.9 eq.), and DIPEA (0.225 mM, 4.5 eq.) in DMF (3 mL) were added to DMF-washed peptide resin (0.05 mM) with agitation for 3 hr. Resin was washed with 20% piperidine in DMF (5 mL) for 2×30 min to remove any excess dye. All manipulations were carried out under foil to exclude light. Peptides were cleaved from the solid support by addition of TFA (9.5 mL), TIPS (0.25 mL), and water (0.25 mL) for 3 hr with shaking at rt. The cleavage solution was reduced to approximately 1 mL under a flow of nitrogen. Crude peptide was precipitated upon addition of ice-cold diethyl ether (40 mL) and recovered via centrifugation. The resulting precipitant was dissolved in 1:1 acetonitrile and water (≈ 15 mL) and lyophilised to yield crude peptide as a solid.

### Peptide purification

Peptides were purified by reverse phase HPLC on a Phenomenex (Macclesfield, UK) Luna C18 stationary phase column (150×10 mm, 5 μM particle size, 100 A pore size) using a preparative JASCO HPLC system. A linear gradient of 20–80% acetonitrile and water was applied over 30 min. Chromatograms were monitored at wavelengths of 220 and 280 nm. The identities of the peptides were confirmed using a MALDI-TOF mass spectrometer using a Bruker ultrafleXtreme II instrument in reflector mode. Peptides were spotted on a ground steel target plate using dihydroxybenzoic acid as the matrix. Peptide purities were determined using a JASCO analytical HPLC system, fitted with a reverse-phase Kinetex C18 analytical column (Phenomenex, 5 μm particle size, 100 Å pore size, 100×4.6 mm). Fractions containing pure peptide were pooled and lyophilised. Peptides were dissolved in buffer (25 mM HEPES pH 7.4 buffer with 150 mM NaCl, 5 mM BME) and their concentrations determined by UV-Vis on a ThermoScientific (Hemel Hempstead, UK) Nanodrop 2000 spectrophotometer using measurement of UV absorbance at 555 nm ($\varepsilon_{555}(\text{TAMRA}) = 85\,000\,\text{mol}^{-1}\,\text{cm}^{-1}$).

### Circular-dichroism spectroscopy

CD data were collected on a JASCO J-810 or J-815 spectropolarimeter fitted with a Peltier temperature controller (Jasco UK). Peptide samples were dissolved at 100 μM concentration in PBS (8.2 mM sodium phosphate, 1.8 mM potassium phosphate, 137 mM sodium chloride, 2.7 mM potassium chloride at pH 7.4). CD spectra were recorded in 1 mm path length quartz cuvettes at 20 °C. The instruments were set with a scan rate of 100 $\text{nm}\,\text{min}^{-1}$, a 1 nm interval, a 1 nm bandwidth and a 1 s response time, and scans are an average of eight scans recorded for the same sample. The spectra were converted from ellipticities (deg) to mean residue ellipticities (MRE; $\text{deg.cm}^2.\text{dmol}^{-1}.\text{res}^{-1}$) by normalising for concentration of peptide bonds and the cell path length using the equation:

$$MRE\left(\text{deg.cm}^2.\text{dmol}^{-1}.\text{res}^{-1}\right) = \frac{\theta \text{X} 100}{c\,X\,l\,X\,b}$$

Where the variable $\theta$ is the measured difference in absorbed circularly polarised light in millidegrees, $c$ is the millimolar concentration of the specimen, $l$ is the path-length of the cuvette in centimetres, and $b$ is the number of amide bonds in the polypeptide, for which the N-terminal acetyl bond was included but not the C-terminal amide.

## Sedimentation-equilibrium analytical ultracentrifugation

Analytical ultracentrifugation (AUC) sedimentation-equilibrium experiments were conducted at 20 °C in a Beckman Optima XL-I analytical ultracentrifuge using an An-60 Ti rotor (Beckman Coulter). Solutions were made up in PBS at 100 µM total peptide concentration. The experiments were run in a two-channel centrepiece. The samples were centrifuged at speeds in the range of 44–60 krpm and scans at each recorded speed were duplicated. Data were fitted to single, ideal species models using SEDFIT (v15.2b)/SEDPHAT, comprising a minimum of four speeds. 95% confidence limits were obtained via Monte Carlo analysis of the obtained fits.

## Fluorescence polarisation

TAMRA-conjugated peptides were diluted to 150 nM and incubated with increasing concentrations of KLC-1/2 TPR protein (typically 0–25 µM) in assay buffer (25 mM HEPES pH 7.4, 5 mM β-mercaptoethanol, and 150 mM NaCl). Measurements were performed on a CLARIOstar (BMG Labtech) microplate reader at room temperature. ΔFP values at each concentration were calculated by subtraction of measurements made without TPR protein. Binding constants ($K_d$) were determined using a one-site specific binding model using GraphPad Prism software. Data were normalised to the calculated $B_{max}$. Where binding was lost or undetectable (e.g. delta Helix1, KinTag-fusion), data were normalised to the $B_{max}$ of the parental protein (e.g. full-length KLC1/2 TPR).

## Cell culture, immunoprecipitation, and fluorescence imaging

HeLa cells obtained directly from ATCC (CCL-2) were grown and maintained in high-glucose Dulbecco's modified Eagle's medium (Sigma-Aldrich) supplemented with 10% fetal bovine serum (Sigma-Aldrich) and 1% penicillin/streptomycin (Gibco) at 37 °C with 5% CO2. Cells were regularly tested for mycoplasma contamination. For immunoprecipitation experiments, cells were transfected with plasmids encoding GFP-MAP7 (Addgene plasmid 46076)(*Metzger et al., 2012*) with HA-KLC2 and HA-KIF5C (*Sanger et al., 2017*). Cell lysis and GFP-TRAP immunoprecipitation experiments were performed as described in *Yip et al., 2016*. For fluorescence imaging, cells were transfected with HA-KIF5C, KLC2-Halo (*Yip et al., 2016*), and GFP-MAP7, and labelled with Halo-TMR ligand as previously described (*Yip et al., 2016*), before fixation and staining with Mouse anti-HA (HA-7) from Sigma-Aldrich, and Alexa 647–conjugated anti-mouse Thermo Fisher Scientific, and imaging on a confocal microscope.

## Acknowledgements

This work was funded by BBSRC grants BB/W005581/1 and BB/Z517276/1. JAC was supported by an EPSRC-funded Doctoral Prize Fellowship. MPD acknowledges support from a Lister Institute of Preventative Medicine Fellowship. CS acknowledges funding by a Wellcome Trust Investigator award (210701/Z/18/Z). JJP and MK acknowledge funding from UKRI Future Leaders Fellowship (MR/T02223X/1). We thank the University of Bristol School of Chemistry Mass Spectrometry Facility for access to the EPSRC-funded Bruker Ultraflex MALDI-TOF instrument (EP/K03927X/1), the BBSRC-funded BrisSynBio centre for access to peptide synthesis and the plate reader (BB/L01386X/1), and the Wolfson Bioimaging Facility for access to this Talos L120C microscope supported by BBSRC equipment grant (BB/X019799/1). We are grateful for the assistance and access to equipment at the GW4 Facility for High-Resolution Electron Cryo-Microscopy, funded by the Wellcome Trust (202904/Z/16/Z and 206181/Z/17/Z). We are grateful to Alex Walker and Bram Mylemans for helpful discussions and to Ferdos Abid Ali for comments on the manuscript.

## Additional information

### Funding

| Funder | Grant reference number | Author |
|---|---|---|
| Biotechnology and Biological Sciences Research Council | BB/Z517276/1 | Derek N Woolfson Mark P Dodding |
| Biotechnology and Biological Sciences Research Council | BB/W005581/1 | Derek N Woolfson Mark P Dodding |
| Wellcome Trust Investigator | 10.35802/210701 | Christiane Schaffitzel |
| UKRI Future Leaders Fellowship | MR/T02223X/1 | Monika Kish Jonathan J Phillips |
| BBSRC equipment | BB/X019799/1 | |
| Wellcome Trust | 10.35802/202904 | |
| Wellcome Trust | 10.35802/206181 | |
| EPSRC-funded Bruker Ultraflex MALDI-TOF instrument | EP/K03927X/1 | |
| BBSRC-funded BrisSynBio centre | BB/L01386X/1 | |
| UKRI Future Leaders Fellowship Renewal | MR/Z000157/1 | Monika Kish Jonathan J Phillips |
| Lister Institute of Preventative Medicine Fellowship | | Mark P Dodding |

The funders had no role in study design, data collection and interpretation, or the decision to submit the work for publication. For the purpose of Open Access, the authors have applied a CC BY public copyright license to any Author Accepted Manuscript version arising from this submission.

### Author contributions

Shivam Shukla, Conceptualization, Formal analysis, Investigation, Methodology, Writing – original draft, Writing – review and editing; Jessica A Cross, Conceptualization, Formal analysis, Investigation, Methodology, Writing – review and editing; Monika Kish, Conceptualization, Formal analysis, Investigation, Writing – review and editing; Sathish KN Yadav, Formal analysis, Investigation, Methodology; Johannes F Weijman, Resources, Investigation, Writing – review and editing; Laura O'Regan, Formal analysis, Investigation, Writing – review and editing; Judith Mantell, Ufuk Borucu, Formal analysis, Methodology, Writing – review and editing; Xiyue Leng, Formal analysis, Investigation, Methodology, Writing – review and editing; Christiane Schaffitzel, Formal analysis, Writing – review and editing; Jonathan J Phillips, Conceptualization, Formal analysis, Supervision, Funding acquisition, Methodology, Writing – original draft, Writing – review and editing; Derek N Woolfson, Conceptualization, Supervision, Funding acquisition, Writing – original draft, Writing – review and editing; Mark P Dodding, Conceptualization, Formal analysis, Supervision, Funding acquisition, Investigation, Methodology, Writing – original draft, Writing – review and editing

### Author ORCIDs

Shivam Shukla ⬤ https://orcid.org/0000-0003-2856-5081
Jessica A Cross ⬤ https://orcid.org/0000-0001-7725-3821
Monika Kish ⬤ https://orcid.org/0000-0002-3661-8641
Johannes F Weijman ⬤ https://orcid.org/0000-0002-2082-5777
Ufuk Borucu ⬤ https://orcid.org/0000-0001-6746-5409
Xiyue Leng ⬤ https://orcid.org/0000-0002-4441-2352
Jonathan J Phillips ⬤ https://orcid.org/0000-0002-5361-9582

Derek N Woolfson [ID] https://orcid.org/0000-0002-0394-3202
Mark P Dodding [ID] https://orcid.org/0000-0001-8091-6534

Reviewer #1 (Public review): https://doi.org/10.7554/eLife.109462.3.sa1
Reviewer #2 (Public review): https://doi.org/10.7554/eLife.109462.3.sa2
Reviewer #3 (Public review): https://doi.org/10.7554/eLife.109462.3.sa3
Author response https://doi.org/10.7554/eLife.109462.3.sa4

## Additional files

### Supplementary files
MDAR checklist

### Data availability
All data needed to interpret, verify and extend the study presented in this article are available in the manuscript, supplementary information and source files.

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
