## [Editor Report · eLife Assessment]

The revised manuscript by Shukla et al. provides **important** mechanistic insights into kinesin-1 autoinhibition and cargo-mediated activation. Through a **convincing** integration of protein engineering, computational modeling, biophysical assays, HDX-MS, and electron microscopy, the study delineates how cargo binding induces an allosteric transition that propagates along the coiled-coil stalk to the motor domains, enhancing MAP7 engagement. The revisions substantially improve clarity, figure annotation, and methodological transparency, leaving the remaining limitations, primarily those inherent to conformational heterogeneity and resolution, appropriately acknowledged. Overall, the updated manuscript presents a coherent mechanism for kinesin-1 activation that will be of broad interest to the motor protein, structural biology, and cell biology communities.

---

## [Referee Report · Reviewer #1 (Public review)]

The authors aim to interrogate the sets of intramolecular interactions that cause kinesin-1 hetero-tetramer autoinhibition and the mechanism by which cargo interactions via the light chain tetratricopeptide repeat domains can initiate motor activation. The molecular mechanisms of kinesin regulation remain a key question with respect to intracellular transport and this study adds important perspectives to our understanding. It has implications for the accuracy and efficiency of motor transport by different motor families, for example the direction of cargos in one or other direction on microtubules.

The authors focus on the response of inactivated kinesin-1 to peptides found in cargos and the cascade of conformational changes that are induced. They also test the effects of the known activator of kinesin-1 - MAP7 - in the context of their model. The study benefits from multiple complementary, albeit relatively low-resolution, methods - structural prediction using AlphaFold3, 2D and 3D analysis of (mainly negative stain) TEM images of several engineered kinesin constructs, biophysical characterisation of the complexes, peptide design, hydrogen/deuterium-exchange mass spectrometry and simple cell-based imaging. Each set of experiments is carefully designed and the intrinsic limitations of each method are offset by other approaches, such that the assembled data convincingly supports the authors' regulatory model of kinesin activation.

This study benefits from prior work by the authors on this system and the tools and constructs they previously accrued, as well as from other recent contributions to the field. This work will be of broad interest to cell and structural biologists, especially those seeking to tackle small and flexible macromolecular complexes, as well as biophysicists and those interested in protein engineering.

---

## [Referee Report · Reviewer #2 (Public review)]

Summary:

In this paper, Shukla, Cross, Kish, and colleagues investigate how binding of a cargo-adaptor mimic (KinTag) to the TPR domains of the kinesin-1 light chain, or disruption of the TPR docking site (TDS) on the kinesin-1 heavy chain, triggers release of the TPR domains from the holoenzyme. This dislocation provides a plausible mechanism for transition out of the autoinhibited lambda-particle toward the open and active conformation of kinesin-1. Using a combination of negative-stain electron microscopy, AlphaFold modeling, biochemical assays, hydrogen-deuterium exchange mass spectrometry (HDX-MS) and other methods, the authors show how TPR undocking propagates conformational changes through the coiled-coil stalk to the motor domains, increasing their mobility, and enhances interactions with the microtubule-bound cofactor MAP7. Together, they propose a model in which the TDS on CC1 of the heavy chain forms a "shoulder" in the compact, autoinhibited state. Cargo-adaptor binding, mimicked here by KinTag, dislodges this shoulder, liberating the motor domains and promoting MAP7 association, driving kinesin-1 activation.

Strengths:

Throughout the study, the authors use clever construct design - e.g. delta-Elbow, ElbowLock, CC-Di and the high-affinity KinTag - to test specific mechanisms by directly perturbing structural contacts or effecting interactions. The proposed mechanism of releasing autoinhibition via adaptor-induced TPR undocking is also interrogated with a number of complementary techniques that converge on a convincing model for activation that can be further tested in future studies.

Weaknesses:

These reflect limits of what the current data can establish rather than flaws in execution. It remains to be tested if the open state of kinesin-1 initiated by TPR undocking is indeed an active state of kinesin-1 capable of processive movement and/or cargo transport. It also remains to be determined what the mechanism of motor domain undocking from the autoinhibited conformation is. But this important study provides the groundwork for testing these open questions.

Comments on revisions:

My original minor concerns have been addressed in the revision.

---

## [Referee Report · Reviewer #3 (Public review)]

Summary:

The manuscript by Shukla and colleagues presents a comprehensive study that addresses a central question in kinesin-1 regulation-how cargo binding to the kinesin light chain (KLC) tetratricopeptide repeat (TPR) domains triggers activation of full-length kinesin-1 (KHC). The authors combine AlphaFold3 modeling, biophysical analysis (fluorescence polarization, hydrogen-deuterium exchange), and electron microscopy to derive a mechanistic model in which the KLC-TPR domains dock onto coiled-coil 1 (CC1) of the KHC to form the "TPR shoulder," stabilizing the autoinhibited (λ-particle) conformation. Binding of a W/Y-acidic cargo motif (KinTag) or deletion of the CC1 docking site (TDS) dislocates this shoulder, liberating the motor domains and enhancing accessibility to cofactors such as MAP7. The results link cargo recognition to allosteric structural transitions and present a unified model of kinesin-1 activation. I recommend acceptance of the manuscript subject to the following additions:

Strengths:

(1) The study addresses a fundamental and long-standing question in kinesin-1 regulation using a multidisciplinary approach that combines structural modeling, quantitative biophysics, and electron microscopy.

(2) The mechanistic model linking cargo-induced dislocation of the TPR shoulder to activation of the motor complex is well supported by both structural and biochemical evidence.

(3) The authors employ elegant protein-engineering strategies (e.g., ElbowLock and ΔTDS constructs) that enable direct testing of model predictions, providing clear mechanistic insight rather than purely correlative data.

(4) The data are internally consistent and align well with previous studies on kinesin-1 regulation and MAP7-mediated activation, strengthening the overall conclusion.

Weaknesses:

(1) While the EM and HDX-MS analyses are informative, the conformational heterogeneity of the complex limits structural resolution, making some aspects of the model (e.g., stoichiometry or symmetry of TPR docking) indirect rather than directly visualized.

(2) The dynamics of KLC-TPR docking and undocking remain incompletely defined; it is unclear whether both TPR domains engage CC1 simultaneously or in an alternating fashion.

(3) The interplay between cargo adaptors and MAP7 is discussed but not experimentally explored, leaving open questions about the sequence and exclusivity of their interactions with CC1.

Comments on revisions:

The authors have addressed my comments satisfactorily.

---

## [Author Response]

The following is the authors’ response to the original reviews

**eLife Assessment**
The manuscript by Shukla et al. provides important mechanistic insights into kinesin-1 autoinhibition and cargo-mediated activation. Using a convincing combination of protein engineering, computational modeling, biophysical assays, HDX-MS, and electron microscopy, the authors reveal how cargo binding induces an allosteric transition that propagates to the motor domains and enhances MAP7 binding. Despite limitations arising from conformational heterogeneity and structural resolution, the study presents a unified mechanism for kinesin-1 activation that will be of broad interest to the motor protein, structural biology, and cell biology communities.

We are grateful for the time and effort from the reviewers and editors in providing fair and constructive comments that have helped to improve the manuscript. Our point-by-point response is provided below.

**Public Reviews:**

**Reviewer #1 (Public review):**
Summary:The authors aim to interrogate the sets of intramolecular interactions that cause kinesin-1 hetero-tetramer autoinhibition and the mechanism by which cargo interactions via the light chain tetratricopeptide repeat domains can initiate motor activation. The molecular mechanisms of kinesin regulation remain an important question with respect to intracellular transport. It has implications for the accuracy and efficiency of motor transport by different motor families, for example, the direction of cargos towards one or other microtubules.Strengths:The authors focus on the response of inactivated kinesin-1 to peptides found in cargos and the cascade of conformational changes that occur. They also test the effects of the known activator of kinesin-1 - MAP7 - in the context of their model. The study benefits from multiple complementary methods - structural prediction using AlphaFold3, 2D and 3D analysis of (mainly negative stain) TEM images of several engineered kinesin constructs, biophysical characterisation of the complexes, peptide design, hydrogen/deuterium-exchange mass spectrometry, and simple cell-based imaging. Each set of experiments is thoughtfully designed, and the intrinsic limitations of each method are offset by other approaches such that the assembled data convincingly support the authors' conclusions. This study benefits from prior work by the authors on this system and the tools and constructs they previously accrued, as well as from other recent contributions to the field.Weaknesses:It is not always straightforward to follow the design logic of a particular set of experiments, with the result that the internal consistency of the data appears unconvincing in places.For example, (i) the Figure 1 AlphaFold3 models do not include motor domains whereas the nearly all of the rest of the data involve constructs with the motor domains;

We appreciate the reviewer’s comment regarding the absence of the motor domains in the AlphaFold3 models shown in Figure 1. These domains were intentionally excluded to improve visual clarity and to better highlight the interaction between the TPR domains and CC1 in the inhibited kinesin-1 conformation. We felt that this simplified presentation in the main figure helps readers focus on the key mechanistic advance introduced in this work at the outset of the paper. For completeness, we have provided full-length kinesin-1 AlphaFold3 models that include the motor domains in the Supplementary Information (Fig. S1), and they are described in detail in the main text. In addition, we have added a note to the Figure 1 legend to explicitly direct readers to these full-length models.

(ii) the kinesin constructs are chemically cross-linked prior to TEM sample preparation - this is clear in the Methods but should be included in the Results text, together with some discussion of how this might influence consistency with other methods where crosslinking was not used.

Thank you. Chemical crosslinking is typically important for obtaining high-quality negative-stain TEM grids of kinesin-1 complexes and has been employed in all prior EM studies by our group and others. While this was described in the Methods, we agree that it should also be stated explicitly in the Results. Accordingly, we have added a sentence to the Results section noting that the proteins were stabilized using the amine-to-amine crosslinker BS3 (“Proteins were also stabilised using the amine-to-amine crosslinker BS3 that was important for achieving reproducibly high-quality samples for imaging.”).

Please see point below for acknowledgement of risks of using crosslinker.

Can those cross-links themselves be used to probe the intramolecular interactions in the molecular populations by mass spec?

We had considered this, however, cross-linking mass spectrometry (XL-MS) has been applied extensively to essentially identical kinesin-1 complexes by Tan et al. (eLife 2023). That work provided important insights into the overall architecture of the complex, including the new head–CC1 interactions. However, as fully acknowledged by the authors, significant ambiguity remained with respect to the positioning of the TPR domains, with many cross-links that could not be straightforwardly rationalized in a single model. These unresolved aspects provided part of the motivation for the present study, as highlighted in the Introduction.

We believe that this ambiguity likely reflects an underlying conformational equilibrium of the kinesin-1 complex (e.g. opening/closing transitions) and/or dynamic docking and undocking of the TPR domains, and lysine-rich features of the TPR domains (most notably the loops that connect the TPR alpha helices) which may make them prone to lock in non-native states, which limits the interpretability of static cross-linking data in this system. In this context therefore, we feel that XL-MS has already been thoroughly explored for kinesin-1 and that its practical limitations in resolving these TPR interactions have been reached.

This consideration was a primary motivation for pursuing cross-linker-free, solution-based approaches, particularly HDX-MS, which we argue provide the most relevant new insights into the assembly and conformational dynamics of the complex. To make this rationale clearer, we have added an explicit note in the HDX-MS section emphasizing that this is a cross-linker-free method. The added text reads:

“To determine how the local structural changes from adaptor binding and shoulder dislocation affected the dynamics of kinesin-1 complexes in solution, as directly and least invasively as possible, and without the risk of cross-linker artefacts.”

In general, the information content of some of the figure panels can also be improved with more annotations e.g. angular relationship between views in Figure 1B, approximate interpretations of the various blobs in Fig 3F, and more thought given to what the reader should extract from the representative micrographs in several figures - inclusion of the raw data is welcome but extraction and magnification of exemplar particles (as is done more effectively in Fig S5) could convey more useful information elsewhere.

We appreciate these suggestions. We have modified the figures throughout the manuscript in line with the reviewer’s points. Raw data is now provided at higher magnification throughout so the reader can better distinguish individual particles, angular relationships have been added and further annotations provided on 2D class averages. We do not want the reader to draw too many conclusions from images of single closed particles (with the exception of open vs closed in Fig S7) as these require averaging and 2D classification to obtain meaningful insights, and so we have not added zoom panels in these cases. Figure 3F has been annotated as requested.

**Reviewer #2 (Public review):**
Summary:In this paper, Shukla, Cross, Kish, and colleagues investigate how binding of a cargo-adaptor mimic (KinTag) to the TPR domains of the kinesin-1 light chain, or disruption of the TPR docking site (TDS) on the kinesin-1 heavy chain, triggers release of the TPR domains from the holoenzyme. This dislocation provides a plausible mechanism for transition out of the autoinhibited lambda-particle toward the open and active conformation of kinesin-1. Using a combination of negative-stain electron microscopy, AlphaFold modeling, biochemical assays, hydrogen-deuterium exchange mass spectrometry (HDX-MS), and other methods, the authors show how TPR undocking propagates conformational changes through the coiled-coil stalk to the motor domains, increasing their mobility and enhancing interactions with the microtubule-bound cofactor MAP7. Together, they propose a model in which the TDS on CC1 of the heavy chain forms a "shoulder" in the compact, autoinhibited state. Cargo-adaptor binding, mimicked here by KinTag, dislodges this shoulder, liberating the motor domains and promoting MAP7 association, driving kinesin-1 activation.Strengths:Throughout the study, the authors use a clever construct design - e.g., delta-Elbow, ElbowLock, CC-Di, and the high-affinity KinTag - to test specific mechanisms by directly perturbing structural contacts or affecting interactions. The proposed mechanism of releasing autoinhibition via adaptor-induced TPR undocking is also interrogated with a number of complementary techniques that converge on a convincing model for activation that can be further tested in future studies. The paper is well-written and easy to follow, though some more attention to figure labels and legends would improve the manuscript (detailed in recommendations for the authors).Weaknesses:These reflect limits of what the current data can establish rather than flaws in execution. It remains to be tested if the open state of kinesin-1 initiated by TPR undocking is indeed an active state of kinesin-1 capable of processive movement and/or cargo transport. It also remains to be determined what the mechanism of motor domain undocking from the autoinhibited conformation is, and perhaps this could have been explored more here. The authors have shown by HDX-MS that the motor domains become more mobile on KinTag binding, but perhaps molecular dynamics would also be useful for modelling how that might occur.

We are grateful for the reviewer’s comments. We agree that the weaknesses the reviewer has outlined define the limitations of the study and establish important priorities for future work, that includes molecular dynamics simulations. An important prerequisite for the latter is a starting model that one has confidence in. We think that our study and earlier work now provide a good experimentally supported foundation for using AF3 generated assemblies for this purpose, by ourselves and others.

**Reviewer #3 (Public review):**
Summary:The manuscript by Shukla and colleagues presents a comprehensive study that addresses a central question in kinesin-1 regulation - how cargo binding to the kinesin light chain (KLC) tetratricopeptide repeat (TPR) domains triggers activation of full-length kinesin-1 (KHC). The authors combine AlphaFold3 modeling, biophysical analysis (fluorescence polarization, hydrogen-deuterium exchange), and electron microscopy to derive a mechanistic model in which the KLC-TPR domains dock onto coiled-coil 1 (CC1) of the KHC to form the "TPR shoulder," stabilizing the autoinhibited (λ-particle) conformation. Binding of a W/Y-acidic cargo motif (KinTag) or deletion of the CC1 docking site (TDS) dislocates this shoulder, liberating the motor domains and enhancing accessibility to cofactors such as MAP7. The results link cargo recognition to allosteric structural transitions and present a unified model of kinesin-1 activation.Strengths:(1) The study addresses a fundamental and long-standing question in kinesin-1 regulation using a multidisciplinary approach that combines structural modeling, quantitative biophysics, and electron microscopy.(2) The mechanistic model linking cargo-induced dislocation of the TPR shoulder to activation of the motor complex is well supported by both structural and biochemical evidence.(3) The authors employ elegant protein-engineering strategies (e.g., ElbowLock and ΔTDS constructs) that enable direct testing of model predictions, providing clear mechanistic insight rather than purely correlative data.(4) The data are internally consistent and align well with previous studies on kinesin-1 regulation and MAP7-mediated activation, strengthening the overall conclusion.Weaknesses:(1) While the EM and HDX-MS analyses are informative, the conformational heterogeneity of the complex limits structural resolution, making some aspects of the model (e.g., stoichiometry or symmetry of TPR docking) indirect rather than directly visualized.

We agree with the reviewers point. Conformational heterogeneity is a significant challenge, and the model has been developed from multiple complementary approaches. A higher resolution cryoEM study remains a priority, but is challenging because of the size, shape and flexibility of the particle, but we hope that some the approaches used here (e.g. nanobody TPR stabilisation, ElbowLock) will provide a path to achieve this.

(2) The dynamics of KLC-TPR docking and undocking remain incompletely defined; it is unclear whether both TPR domains engage CC1 simultaneously or in an alternating fashion.

We agree that this is a limitation. We strongly suspect that the TPR domains dynamic and are working to overcome experimental challenges to resolve this important outstanding question. We have expanded the discussion section to better highlight this important priority.

(3) The interplay between cargo adaptors and MAP7 is discussed but not experimentally explored, leaving open questions about the sequence and exclusivity of their interactions with CC1.

We agree that this is a limitation but will be an important priority for future studies.

**Recommendations for the authors:**

**Reviewer #1 (Recommendations for the authors):**
There are a number of places where the text could be more precise or clear, or the figures could be designed to be more informative:(1) The word "unitarily" is used in several places, and I don't know what it means in this context.

We have changed the phrasing throughout the manuscript to this term. We were attempting to contrast with presumed cooperative multivalent interactions in the context of the kinesin-1 tetramer but agree that this choice of word doesn’t quite achieve that.

(2) On page 5 the phrase "We focused on the ElbowLock background" is introduced and needs to be explained more clearly.

Thank you. We have amended the text to read “This KIF5C construct contains a short 5 amino acid deletion that restricts flexibility around the elbow and helps maintain particles in their lambda conformation, providing homogenous samples, and facilitating subsequent analysis (34).”

(3) On page 6, the phrase "To improve the resolution of our images, we turned to single-particle cryoEM analysis" is imprecise - what do the authors mean by the resolution of the images? Cryo-EM data does not always guarantee a higher resolution structure, but it offers the possibility of visualising finer structural features. This is probably what is meant here, but needs to be stated more precisely.

We have amended the text to ‘visualise finer structural details’ as suggested.

(4) Page 7 - "suggesting that TPR domains had loosely dissociated from the core" - I don't think the evidence points to dissociation of KLCs from the complex, but the phrase "loosely dissociated" implies this - would benefit from rephrasing.

We have changed this to ‘undocked’ for consistency with other descriptions in the manuscript.

(5) Was the effect of the CC-Di insertion (ΔTDS) detectable by AlphaFold prediction? It would be interesting to include this, partly for completeness and partly because a slightly imperfect and maybe a more dynamic coiled-coil in this region of the molecule may be important in supporting the conformational changes required for activation.

Thank you for this suggestion. Modelling of deltaTDS complex indeed shows displacement of the TPR domains. In the standard 5 output models, the TPR domains now occupy a variety of different positions, all with essentially zero confidence (high position error). Consistent with biochemical data, the CCDi insertion is modelled with with no overall disruption to the architecture or length of CC1 as expected. We think that this is a valuable addition to the study and have included it as a new supplementary figure (Fig S5), with main text reading.

…. “Supporting this, models of ΔTDS complexes using AF3 showed the expected seamless insertion of CCDi into CC1, with displacement of the TPR domains to a variety of different positions, in 5 models, all with high position error with respect to KHC (Fig S5).”

(6) Figure S1 has two sections designated (C) in the legend.

Corrected

(7) Figure S3 - given the resolution and level of interpretation of the 3D reconstructions, it is not relevant to include an FSC curve, but other standard information, such as angular distribution and any evidence of variability from 3D classifications (and how many particles per 3D class) should be included for all structures.

Thank you, a complete workflow for all complexes has now been provided in Figure S8 with the information requested. In each case there were typically two ‘good’ classes. For ElbowLock, this included one without a prominent shoulder, consistent with 2D classification and quantification. We assume this may reflect a docking/undocking equilibrium. For the deltaTDS and KinTag particles, neither class showed the shoulder feature. The main text has been modified to reflect this and reads “For ElbowLock complexes, this resulted in classes with and without a prominent shoulder, in agreement with 2D classification. For ElbowLock-ΔTDS and ElbowLock-KinTag complexes, no prominent shoulder containing classes were observed.”

**Reviewer #2 (Recommendations for the authors):**
Overall, the figures would benefit from more labels for clarity, some examples and suggestions below:(1) Figure 1A - Connect motors to the rest of the structure e.g., wiggly lines.

Corrected.

(2) Figure 1B - Add arrows and angles to indicate different views of the model.

Corrected.

(3) Figure 1B - Label TPR1-6 (e.g., inset zoom in).

Corrected.

(4) Figure 2D and 3D - Label the lack of a shoulder in all averages (perhaps with an arrow instead of a circle to not obscure density), include an example average which shows prominent shoulder density.

Corrected. Full sets of classes showing shoulder like features for deltaTDS and KinTag complexes are now shown in Figure S4.

(5) Figure 3D: Label motor domains and elbow as in other figures.

Corrected.

(6) Methods: Include more information on how EM classes were compared to AF projections (e.g., Figure 1D). Was this done visually or computationally? Likewise, more information is needed on how classes were judged to have prominent/weak shoulder density (Figure 2D). In the figure legend, there is a statement that "Full sets of classes are provided in Fig. S4" but this is absent in the supplement.

Thank you. This information has been added to the methods.

“For comparison to the AF3 model, simulated density was generated using the molmap command in ChimeraX (73) filtering to 15 Å, and projections were generated/selected automatically using the Reference Based Auto Selected 2D function in CryoSPARC”.

Full sets of classes are now provided in Figure S4.

(7) Figure 1-3 - Raw micrographs are a very useful inclusion but would benefit from being a more zoomed-in view (e.g., Figure S5 scale). Particularly useful for 3C, where the mixture of open and closed would be good to see.

Higher zoom micrographs have been provided throughout.

(8) Figure 5D: Panels too small to see the result, suggest making full width and moving E below.

Thank you. We have expanded the panel and moved the model to a new Figure 6.

(9) Figure S1: PAE plot convincing, but pLDDT colour models needed.

A representative model coloured for pLDDT has been added to Figure S1. Most of the structure sits within the light blue confident range (90 > pLDDT > 70) with the exception of the disordered regions and neck coil.

(10) Figure 5B: Reason for the variable inputs?

The reviewer raises an interesting point. The slightly reduced expression of deltaElbow and slightly increased expression of ElbowLock is a consistent feature of these experiments. We note that this effect is in the ‘opposite direction’ to the impact on binding to MAP7 and so does not affect our conclusions from the experiment. However, we wonder whether opening and closing of the complex may impact on turnover of kinesin proteins, which could have implications for their normal homeostasis and possible degradation after transport in polarised cells. We are considering how to explore this going forwards. We have added a note to the results section to highlight this interesting observation to the reader.

“We also noted slightly elevated expression of ElbowLock complexes and slightly lower expression of DeltaElbow complexes, suggesting that opening/closing of the complex could impact on kinesin-1 turnover”

(11) Figure legend 5B: Insufficient detail, the end result is stated, but the three separate gels are not described.

Legend has been expanded.

(12) Figure 3F: Currently somewhat problematic. It is unclear if the models are in the same view, and so comparison is difficult. Figure 1C (bottom right) shows class averages with a clear, separate CC density, so the relatively featureless model in this region is puzzling. A statement on how the three model views are related to each other, if aligned with each other, would be useful.

We appreciate the reviewers point. Models were aligned in Chimera, using the fit in map command. Because of the limited features of the models presumably due to flexibility, achieving a good alignment for all three models was challenging, but we think that showing the 180-degree rotations is probably about the best we can achieve here.

(13) The following statement is too strong: "Nonetheless, we obtained reference-free 2D class averages that appeared to show full-length 'side' views of the complex with clear definition of the elbow, hinge 2, and KHC-KLC (coiled-coil) interface features which enabled us to identify CC1 confidently (Fig. 1D)". Given that the negative-stain EM data were collected primarily to validate the AlphaFold model, the assignment of CC1 should be described as consistent with rather than confidently identified from the class averages. The resolution of the EM data does not independently support such an assignment, and the wording needs to be softened.

We appreciate the reviewer’s point, we have softened the wording as suggested. The paragraph now reads.

“To visualise finer structural details, we turned to single-particle cryoEM analysis of frozen-hydrated samples. We were unable to obtain optimal samples suitable for determining the complete structure. Nonetheless, we obtained reference-free 2D class averages that appeared to show full-length ‘side’ views of the complex with clear definition of the elbow, hinge 2, and KHC-KLC (coiled-coil) interface features (Fig. 1D). The motor domains were poorly resolved in these classes, suggesting that the head assembly is somewhat flexible relative to the coiled coil/TPR body. A comparison to low-pass filtered back-projections from the AF3 model (without motor domains) revealed density at a position concurrent with the docked TPR domains (Fig. 1D).”

(14) There is a typo in the figure legend of Figure 3 - (E) and (F) should be (F) and (G).

Corrected

**Reviewer #3 (Recommendations for the authors):**
I recommend the following additions:(1) Figure 1 labeling - In panel A, please label the "linker domain" and the "KLC subunits" explicitly to help orient the reader. In panel B, please mark the "TPR shoulder" corresponding to the docked TPR domains on CC1; this will help the reader connect parts B and C.

Thank you, we have modified Figure 1A with this additional information.

(2) The TPR docking site (TDS) is a central structural element, and its sequence boundaries are provided in the Methods. It would help to visualize this directly in Figure 2A or in an inset.

We hope that the reviewer agrees that the zoomed in model in Figure 5A (alongside MAP7) provides a sufficiently detailed view of the structural interface to highlight the orientation of TPR1 with respect to CC1. The side chain contacts in the model are very plausible and confidently predicted (and can be straightforwardly reproduced in AF3 using the sequence information provided in the methods), but as our study has not explored this interaction at the single residue level, we would prefer not to imply this to the reader at this stage.

(3) The authors' model of cargo-induced TPR dislocation is convincing. However, the Discussion could benefit from a clarification on whether both KLC-TPR domains are expected to be bound simultaneously or if a dynamic exchange occurs, as the EM data suggest potential asymmetry.

Thank you, please see point 5 below where we have modified the discussion to reflect the reviewer’s thoughtful comments.

(4) The HDX-MS analysis is comprehensive, but the authors may want to briefly comment on the coverage of low-signal regions (especially within CC2-CC3) to enhance clarity.

We have added an additional supplementary figure (S10) showing sequence coverage. Overall, this is 88% but with some lower coverage around KHC-CC0 (neck) and the acidic linker that connects the KLC coiled-coil to the TPR. We have added a note to the main text to reflect this.

“Sequence coverage was high (overall 88%) with the exception of KHC-CC0 (neck coil) and the acidic-linker region that connects the KLC coiled-coil to the TPR domains where coverage was lower”

(5) In the Discussion, the proposed interplay between MAP7 and cargo adaptors is intriguing, especially considering the results from Anna Akhmanova's lab showing that MAP7 activates kinesin-1 processivity. Do the authors suggest that competition for CC1 is mutually exclusive or sequential? The answer has mechanistic implications.

We have been considering questions for some time, and the short answer is that we don’t fully understand the dynamics yet. However, we appreciate the reviewer’s prompt to clarify our thinking on this. We have attempted to do this in a revised discussion section where we more explicitly outline these outstanding questions.